# TORC2 inhibition triggers yeast chromosome fragmentation through misregulated Base Excision Repair of clustered oxidation events

Kenji Shimada [1], Cleo V. D. Tarashev[1,6], Stephanie Bregenhorn[2], Christian B. Gerhold [1,3], Barbara van Loon [4], Gregory Roth [1], Verena Hurst [1], Josef Jiricny[2], Stephen B. Helliwell [5,7] & Susan M. Gasser [1,8] ✉

Combinational therapies provoking cell death are of major interest in oncology. Combining TORC2 kinase inhibition with the radiomimetic drug Zeocin results in a rapid accumulation of double-strand breaks (DSB) in the budding yeast genome. This lethal Yeast Chromosome Shattering (YCS) requires conserved enzymes of base excision repair. YCS can be attenuated by eliminating three *N*-glycosylases or endonucleases Apn1/Apn2 and Rad1, which act to convert oxidized bases into abasic sites and single-strand nicks. Adjacent lesions must be repaired in a step-wise fashion to avoid generating DSBs. Artificially increasing nuclear actin by destabilizing cytoplasmic actin filaments or by expressing a nuclear export-deficient actin interferes with this step-wise repair and generates DSBs, while mutants that impair DNA polymerase processivity reduce them. Repair factors that bind actin include Apn1, RFA and the actin-dependent chromatin remodeler INO80C. During YCS, increased INO80C activity could enhance both DNA polymerase processivity and repair factor access to convert clustered lesions into DSBs.

Human cells are exposed continuously to oxidative damage, which arises both from endogenous metabolic activity and exogenous sources. These events lead to DNA adducts or single-strand (ss) nicks at a very high rate, i.e., more than $10^3$ events per mammalian genome per day[1]. The conserved DNA damage repair pathways of short- and long-patch base excision repair (BER) guarantee genome integrity against base oxidation. In specialized cases, DNA base lesions can be exploited in a controlled manner, for example, in myeloid cells undergoing differentiation or antibody diversification[2], yet the spontaneous and random DNA base oxidation caused by Reactive Oxygen Species (or ROS) is mutagenic if not efficiently repaired[1]. Indeed, a common oxidized base, 7,8-dihydro-8-oxo-guanine (8-oxo-G), can trigger the C:G to A:T transversions, creating a mutational burden found in lung, breast, ovarian, gastric, and colorectal cancers[3].

BER is a multistep process generally initiated by DNA *N*-glycosylases[3,4] that helps cells cope with DNA oxidation and is

[1]Friedrich Miescher Institute for Biomedical Research, Fabrikstrasse 24, Basel, Switzerland. [2]Institute of Molecular Life Sciences, University of Zürich, Winterthurerstrasse 190, 8057 Zürich, Switzerland; and Institute of Molecular Cancer Research, University of Zurich, Zurich, Switzerland. [3]BÜHLMANN Laboratories AG, Baselstrasse 55, Schönenbuch, Switzerland. [4]Norwegian University of Science and Technology; Department of Clinical and Molecular Medicine, Erling Skjalgssonsgatan, Trondheim, Norway. [5]Novartis Institutes of Biomedical Research, Novartis Intl. AG, Basel, Switzerland. [6]Present address: Dynamics Group AG., Av. de Rumine 5, Lausanne, Switzerland. [7]Present address: Cellvie AG, Zurich, Switzerland. [8]Present address: University of Lausanne, Department of Fundamental Microbiology, and Agora Cancer Center, ISREC Foundation, rue du Bugnon 25A, Lausanne, Switzerland. ✉e-mail: susan.gasser@fmi.ch

conserved across all eukaryotic species. Monofunctional glycosylases recognize and remove damaged DNA bases, generating apurinic or apyrimidinic (AP) sites. Bifunctional *N*-glycosylases possess an additional DNA lyase activity that cleaves the DNA backbone by means of a β-elimination reaction, generating a ss break (SSB) with a 3′-deoxyribose-phosphate (3′-dRP) (reviewed in ref. 5). Examples of bifunctional *N*-glycosylases in budding yeast are Ogg1, Ntg1 and Ntg2 (OGG1, NTH1 and NTH2 in mammals[6,7]), although human OGG1 also can also be monofunctional, and simply remove the damaged base[8]. AP sites and 3′-dRPs are then processed by AP endonucleases (Apn1/APE1, Apn2/APE2) to produce SSBs with a 3′-OH, providing a suitable substrate for DNA polymerase (pol)-mediated elongation[5].

Two BER pathways can repair Apn1/APE1-generated SSBs. The dominant process in mammalian cells called short-patch BER (SP-BER), makes use of DNA Pol β together with XRCC1 and DNA ligase III (*LIG3*). A second pathway, called long-patch BER (LP-BER), makes use of replicative DNA polymerases ε and δ[9,10], which are able to fill in larger gaps on the damaged strand, after which the residual nick is sealed by DNA ligase I (Cdc9/LIG1). Besides DNA Pol ε and δ, budding yeast has two Pol β-like enzymes: Pol4, which resembles Pol λ, and Trf4 (also known as Pap2), which has a 2-deoxyribose-5-phosphate lyase activity, possibly implicated in SP-BER[11]. However, there is no homolog of XRCC1, PARP1, or ligase III in *S. cerevisiae*, leaving LP-BER the dominant base excision repair pathway in this organism[12,13]. A less well-characterized variant of LP-BER in yeast, called nucleotide incision repair or NIR[14,15], bypasses the action of the glycosylase and initiates repair following AP endonuclease cleavage.

During BER, SSBs arise as an intermediate in the multi-step repair process. Yet SSBs pose a serious risk to genome replication or repair because polymerases encountering a nick on the nontemplate strand can generate DSBs[16,17]. To avoid DSBs, strand-cleavage and gap-filling activities of BER must be coordinated with replication. Moreover, when nucleotide adducts are found adjacent to each other on opposite strands, adjacent lesions must be processed sequentially[18,19]. Indeed, "abortive" BER and Zeocin overdose have been shown to generate DSBs after exposure to radiation (γIR) or the radiomimetic antibiotics Zeocin or bleomycin, both in bacteria[20,21] and in human cells[22,23]. Yet it has remained unclear what mechanisms avoid this and ensure the sequential repair of closely juxtaposed base lesions[18,19].

We previously showed that the depolymerization of filamentous actin by inhibition of TORC2, but not of TORC1, can drive a rapid fragmentation of the yeast genome, called YCS if cells are exposed to relatively low levels of Zeocin, bleomycin, or γIR[24]. These agents generate oxidized base damage, SSBs, and DSBs in decreasing abundance[24,25]. In YCS, TORC2 inhibition allows the rapid accumulation of randomly distributed DSBs at roughly 100 kb spacing, which leads to cell death[24]. The multisubunit TORC2 complex is not found in the nucleus of yeast cells, and TORC2-induced YCS can be mimicked by inhibiting its two downstream effector kinases, Ypk1 and Ypk2, homologs of the mammalian SGK1[26]. These control actin polymerization, sphingolipid synthesis, and turnover of the major phospholipid, phosphotidylcholine[27]. While there is no evidence to date that phospholipid homeostasis has a role in YCS, we found that the destabilization of actin filaments by Latrunculin A (LatA) also mimics TORC2 or Ypk1/Ypk2 inhibition in yeast cells that are exposed to Zeocin[24]. Here, we show that elevating nuclear levels of actin by expressing an actin mutant that lacks nuclear export signals, has the same effect.

Nuclear actin has been implicated in various aspects of DNA damage repair in fly and mammalian cells[28,29]. Actin binding factors, such as mammalian WASP and Arp2/Arp3, are thought to contribute to DSB repair by forming repair foci[30,31], or by helping replication fork restart through the sequestration of stalled forks away from PrimPOL, a promiscuous DNA polymerase that both primes and elongates[32,33]. Nuclear actin has also been proposed to act by stabilizing RPA, the ss-DNA binding complex[34,35]. Under some conditions, short transient nuclear actin filaments have been detected in mammalian nuclei, although these are generally detected by expressing a high-affinity actin-binding domain fused to a nuclear localization signal (LifeACT®), which can generate artefacts by sequestering actin in the nucleus[29]. In fact, most nuclear actin exists as a globular (G) form in complex with actin-related protein Arp4 (BAF53 in humans), and the heterodimer is an integral component of multiple large nucleosome modifying complexes, notably INO80C, SWR/SRCAP, TIP60 and BAF1/BRG1[36,37]. We note that actin filaments have not been detected in the budding yeast nucleus[29] and that the actin filaments found in mammalian nuclei are generally transient, lasting < 20 s[32,38].

Here, we investigated in detail the DNA-associated events that drive chromosome breakage when TORC2 inhibition is combined with sublethal doses of Zeocin. The resulting DSB formation is shown to be dependent on the BER machinery, notably AP endonucleases, and on an enhanced processivity of DNA Pol δ and ε. Based on earlier data showing that DNA polymerase processivity is upregulated by INO80C[39–42] and that DNA accessibility increases upon TORC2 inhibition, we propose that nuclear actin simulates the processing of adjacent or clustered lesions to generate DSBs, and we identify enzymes that may be involved in aberrant LP-BER. This highlights an unexpected link between cytoplasmic actin filament integrity and BER and may help us understand the role of oxidative damage in combinatorial cancer therapies.

## Results

### TORC2 inhibitors and Zeocin shatter chromosomes without cell cycle progression

A chemicogenetic screen identified a TORC1 and TORC2 inhibitor, NVP-BHS345 (hereafter BHS), as being synthetic lethal in budding yeast exposed to sublethal doses of the radiomimetic chemical Zeocin or γ-irradiation[24]. The combination of Tor kinase inhibition and Zeocin-induced DNA damage led to the very rapid conversion of the 16 full-length chromosomes, which are readily visualized by non-denaturing Clamped Homogeneous Electric Field (CHEF) gel electrophoresis, into fragments of 50–200 kb[24] (Fig. 1a). We identified a closely related imidazoquinoline, CMB4563[43] (hereafter called CMB), which is 10- to 20-fold more potent in the YCS assay than BHS (Supplementary Fig. 1a, b). Using haploinsufficiency profiling (HIP[44]), we again confirmed the drug target as the TORC2 complex (Supplementary Fig. 1c). The budding yeast TORC2 activity depends on the PIKK-like kinase Tor2, a protein closely related to Tor1, which is found exclusively in the TORC1 complex[45]. Importantly, YCS is not provoked by inhibition of TORC1 by rapamycin, and a point mutation in the catalytic pocket of Tor2 conferred resistance to Zeocin-induced YCS in the presence of either BHS[24] or CMB[46]. This confirmed that the on-target inhibition of TORC2 triggers massive chromosome breakage in the presence of Zeocin-induced damage. Here we use the two TORC2 inhibitors interchangeably albeit at different concentrations.

Zeocin, a member of the bleomycin family of antibiotics, causes base oxidation and AP sites when incubated with DNA[25,47], and generates SSBs and DSBs in a 10:1 ratio[25]. The rare DSBs were proposed to arise from clustered lesions on opposite DNA strands[25,47]. Because we do not detect a general increase in abasic sites upon inhibition of TORC2 (Supplementary Fig. 2a), we suspected that genome fragmentation in YCS arises from the misregulation of one or more repair pathways, rather than from increased uptake or enhanced action of Zeocin. This conclusion is reinforced by the fact that γIR can replace Zeocin in this assay[24]. Importantly, the elimination of the two main DSB repair pathways, that of homologous recombination (HR) and non-homologous end-joining (NHEJ), by deleting Rad51 and Ligase IV, neither mimicked nor suppressed the YCS phenotype[24] (Fig. 1a). Thus, YCS does not arise from blocking DSB repair.

To see if other types of DNA damage trigger YCS, we challenged an exponentially growing yeast culture with $H_2O_2$ and methyl

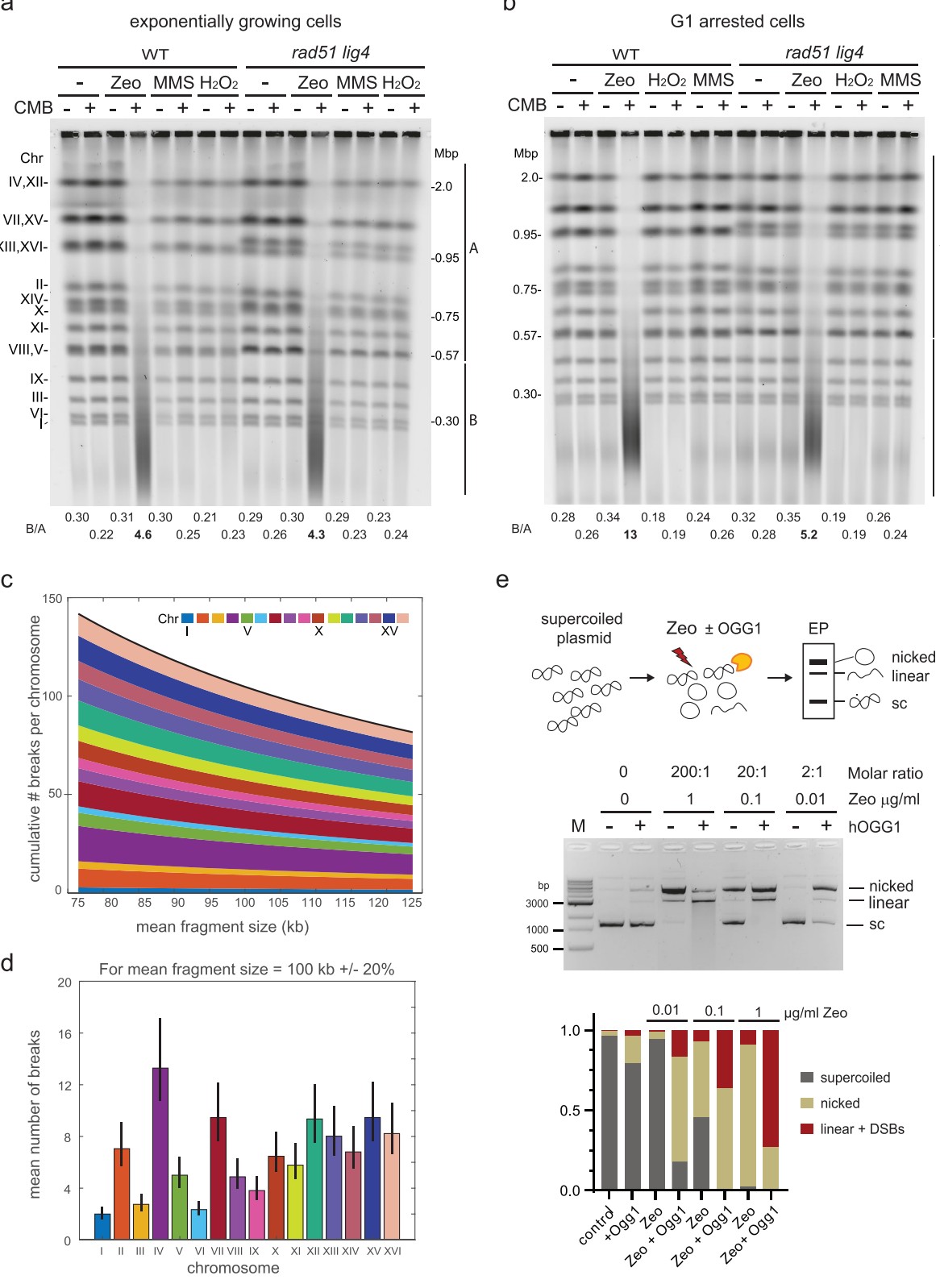

**c** cumulative # breaks per chromosome; Chr I–XVI

**d** For mean fragment size = 100 kb +/- 20%

**e** supercoiled plasmid → Zeo ± OGG1 → EP → nicked / linear / sc

methanesulfonate (MMS) with or without CMB and compared these with Zeocin. $H_2O_2$ and MMS cause random DNA oxidation and base alkylation, respectively, while Zeocin, like bleomycin, tends to induce paired lesions on opposite strands[25,48]. Here, we carefully avoided heat-induced cleavage of MMS-induced lesions by isolating yeast DNA at 30 °C[49] and found that neither MMS-induced alkylation nor oxidation by $H_2O_2$ led to genome breakage upon TORC2 inhibition (Fig. 1a), which again was independent of HR and/or NHEJ repair.

Passage through the S phase can enhance the generation of DSBs if replication forks encounter unrepaired SSBs. We therefore tested whether cells arrested in the G1 phase by pheromone treatment would be shielded from YCS. However, chromosome breakage occurred similarly in G1-arrested cells as in exponentially growing cultures (Fig. 1b), ruling out a requirement for genomic replication in YCS (for FACS profiles of arrested and growing cells, see Supplementary Figs. 2b and 5). We conclude that chromosome breakage upon TORC2

**Fig. 1 | TORC2 inhibition and Ogg1 activity convert Zeocin-generated DNA base oxidation into DSB without DSB repair machinery or replication. a, b** Massive yeast chromosome fragmentation occurs in G1-arrested as well as exponential cultures. Wild-type (WT, GA-1981) and an isogenic *rad51 lig4* double mutant (GA-6098) were grown exponentially (**a**), or arrested in G1 by α-factor for 1.5 h (**b**). Cells were treated with 50 μg/ml Zeocin, 1 mM H$_2$O$_2$, or 0.05% MMS alone, or in combination with the potent TORC2 inhibitor CMB4563[43] (CMB; Supplementary Fig. 1a) at 0.5 μM for 80 min. Genomic DNA was isolated in an agarose plug at 30 °C as described in Methods, chromosomes were separated on a non-denaturing CHEF gel and stained with SYBR safe. The signal intensity of each lane was measured, and ratios of the background-depleted signal of B (from 0.56 Mbp down to ~20 kbp) over A (above 0.57 Mbp, or from the Chr VIII/V band through largest Chr IV/XII) are indicated below the image (data summarized in Supplementary Data 1; and FACS analyses in Supplementary Fig. 5). **c** To obtain a given mean fragment size (*x*-axis), the mean

number of randomly occurring breaks for each chromosome was calculated and plotted along the *y*-axis from Chr I to Chr XVI, as described in Supplementary Fig. 2, legend. The black line represents the number of DSBs per genome needed to generate the indicated fragment size. **d** Theoretical mean number of randomly distributed DSB per chromosome for a mean fragment size of 100 kb was determined by 1000 repeat calculations as in panel **c**. The black bars indicate a 20% error in the estimated mean fragment size (measured mean fragment size ranges from 80 to 120 kb under standard conditions). **e** Zeocin treatment induces substrate for hOGG1-mediated linearization of supercoiled plasmids in vitro. Supercoiled plasmid DNA was mixed with the indicated concentration of Zeocin for 60 min at 37 °C, in the indicated molar ratio, and then incubated with or without purified hOGG1 for 30 min at 37 °C. Plasmid DNA was recovered and analyzed on an agarose gel to separate nicked, linearized, and supercoiled forms. The signal intensity of plasmid forms was quantified by ImageJ, and their ratio was plotted below the gel.

inhibition is quite selective for Zeocin-type base oxidation or abasic site induction. It occurs rapidly (within 30 min[24]), is irreversible, does not arise from DSB repair, nor does it require S phase passage (Fig. 1b).

To understand the frequency of DSBs incurred during YCS, we developed a means to quantify the damage. The integrity of the 16 yeast chromosomes is monitored on nondenaturing CHEF agarose gels followed by staining with dyes that detect dsDNA at least 10-fold more efficiently than ssDNA (Fig. 1a). To quantify and compare degrees of chromosome fragmentation, we scan dsDNA intensity vertically lane by lane after CHEF gel separation, and calculate the intensity of DNA signal > 0.57 Mbp (which includes intact Chr V and VII) vs fragmented chromosomal DNA (< 0.56 Mbp, which includes the four smallest chromosomes intact). This yields a normalized internal ratio that increases with YCS efficiency (B/A, Methods and Supplementary Fig. 2c). This ratio is particularly robust when monitoring relative chromosome integrity within a single experiment on one gel after background subtraction.

The average fragment size resulting from extensive fragmentation (YCS conditions of 50–80 μg/ml Zeocin for 60–80 min) is 100 ± 20 kb. To convert all 16 chromosomes of budding yeast into fragments of 100 ± 20 kb requires 112 ± 22 DSBs per genome, assuming a random or unbiased spatial distribution of the lesions (Fig. 1c, d and Supplementary Fig. 2c); similarly, a mean fragment size of 200–300 kb corresponds to a mean number of DSB of 25–50 genome-wide ("Methods"). Based on Povirk's calculation that the ds:ss break ratio for bleomycin (closely related to Zeocin) is 1:10[25], we estimate that the level of Zeocin we standardly use is generating over 1000 modified bases per genome. Nonetheless, given the absence of chromosome breakage in wild-type strains, we can assume that this level of insult is readily repaired under normal growth conditions. In fact, budding yeast viability is only slightly impaired even at much higher doses of Zeocin[50]. Despite a remarkable capacity for BER under unperturbed growth, we estimate that upon TORC2 inhibition, 50-80 μg/ml Zeocin causes up to 130 DSBs, such that even low levels of Zeocin are synthetically lethal[24].

The clustering of Zeocin-induced lesions is thought to stem from the production of a hydroxyl radical that causes DNA base oxidation near a Zeocin-induced nick[25,47,51]. Rapid processing of clustered damage by an enzyme that cleaves the sugar backbone could, therefore, generate a DSB. We checked the impact of Zeocin on a supercoiled (sc) plasmid, to see first whether it does generate 7,8-dihydro-8-oxo-guanine (8-oxo-G) in vitro, and, secondly, to monitor whether a plasmid exposed to Zeocin serves as substrate for OGG1, the bifunctional N-glycosylase that can both generate an abasic site at 8-oxo-G and cleave DNA by means of a β-elimination reaction with its lyase activity[52]. We incubated 100 ng (about 2 nM) of supercoiled DNA plasmid exposed to different amounts of Zeocin with purified human OGG1 and monitored the conversion of the sc plasmid to a relaxed (nicked) or linearized state. Without Zeocin, the relaxation of the sc template was very low, reflecting the rare occurrence of spontaneous

8-oxo-G. After treatment with a 2:1 molar ratio of Zeocin (0.01 μg/ml), the plasmid remained supercoiled in the absence of OGG1, but its addition converted nearly all of the Zeocin-treated plasmid into a relaxed or nicked circular form (Fig. 1e). Although at this concentration (0.01 μg/ml Zeocin), OGG1 did not induce DSBs, at 10 times higher concentrations (0.1 μg/ml Zeocin), OGG1 converted 40% of the total template to linear DNA, indicative of DSB induction (Fig. 1e). Not unexpectedly, very high Zeocin to template ratios (200:1), led to all sc plasmid being nicked and/or linearized (Fig. 1e).

We draw two conclusions from this in vitro analysis. First, Zeocin at a 200-fold molar excess can efficiently relax and inefficiently linearize a sc plasmid without added enzyme, consistent with the proposal that it introduces juxtaposed lesions. Secondly, after treatment with 10 to 100-fold lower levels of Zeocin, a circular substrate can be converted efficiently into a nicked or linear product by OGG1, due to break induction. Given that OGG1 is the first enzyme in the BER pathway (Fig. 2a), we next examined whether BER drives YCS in vivo.

## BER glycosylases and AP nucleases are implicated in YCS
There are three *N*-glycosylases that initiate oxidative DNA base repair in yeast, Ogg1, Ntg1, and Ntg2[3] (see BER scheme, Fig. 2a). Because the triple knockout is expected to be highly mutagenic, we created a *ntg1Δ ntg2Δ* double deletion strain bearing a degron-tagged OGG1 gene (*OGG1*-deg *ntg1Δ ntg2Δ*; called NGΔ). The addition of the auxin Indole-3-acetic acid (IAA) led to the rapid degradation of Ogg1 (Supplementary Fig. 3a), thereby eliminating all glycosylase activity. To NGΔ and isogenic wild-type cultures we added 0.5 mM IAA prior to the induction of damage by the addition of Zeocin with or without the TORC2 inhibitor BHS (Fig. 2b). In the absence of BHS, 75 μg/ml Zeocin only weakly affected yeast chromosome integrity (B/A = 0.42 vs 0.33), as did the highest level of BHS alone (B/A = 0.39; Fig. 2b). When combined, we observed dose-dependent chromosome shattering, as expected, with full fragmentation at 10 μM BHS and 75 μg/ml Zeocin (B/A > 5.0; mean fragment size < 300 kb, or at least 25 DSBs). Importantly, the glycosylase-deficient NGΔ strain showed strongly reduced chromosome shattering under identical conditions (B/A = 1.7; Fig. 2b). Although loss of the glycosylases did not eliminate all DSB formation, this result suggested that YCS does depend on the action of BER enzymes to generate breaks. Given that the loss of Ogg1 alone (*ogg1Δ*) supported wild-type YCS (Supplementary Fig. 3b), we conclude that the three yeast glycosylases, Ogg1, Ntg1 and Ntg2, act redundantly in this context.

## Loss of AP endonuclease activity results in strong resistance to YCS
We next asked if the ablation of enzymes that act downstream of the glycosylases shows a similar resistance to YCS. Notably, the AP endonucleases are thought to cleave the DNA backbone at abasic sites created by glycosylases in BER[53], but might also act directly on Zeocin-induced lesions. Yeast Apn1 and Apn2 possess canonical AP

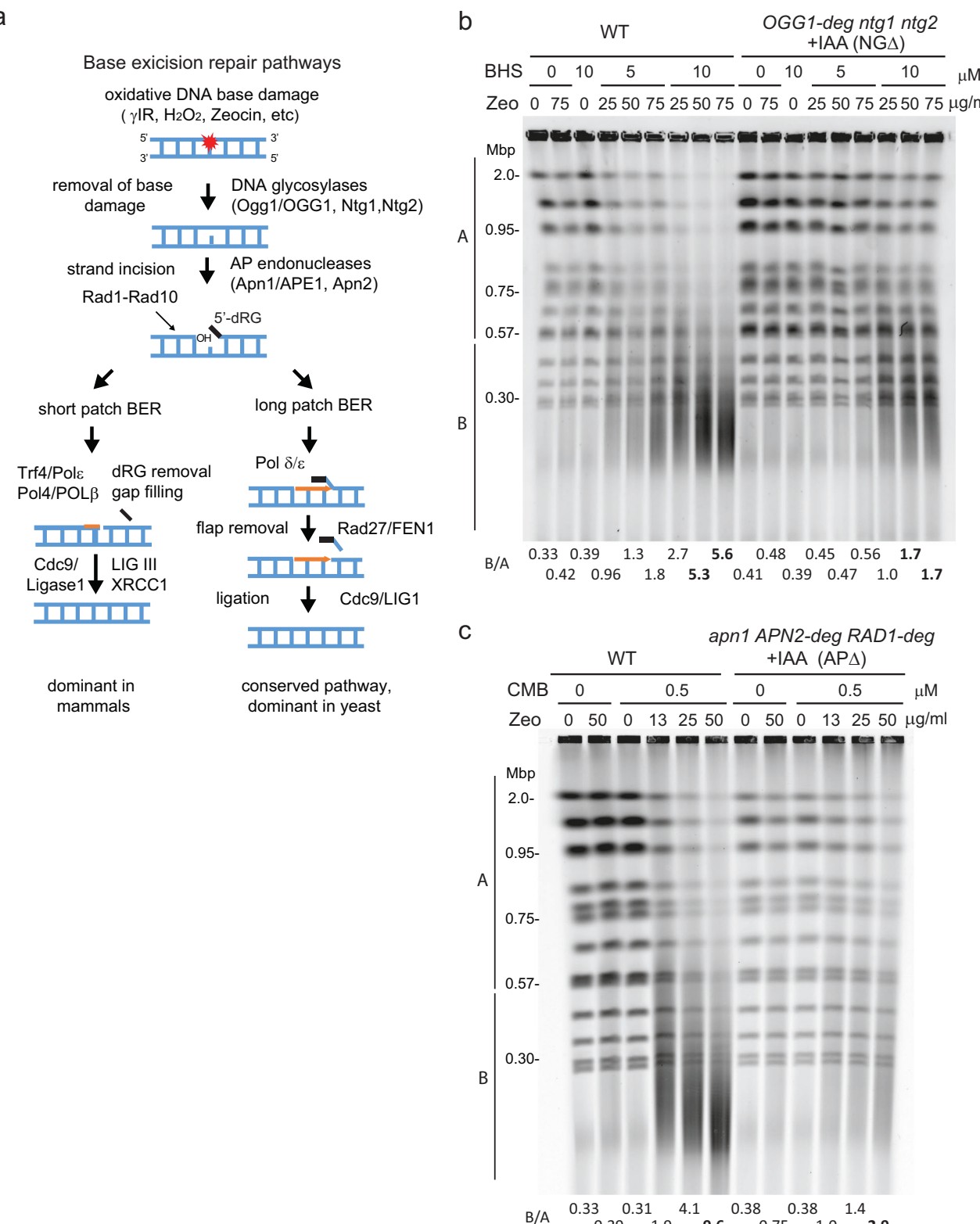

**Fig. 2 | BER *N*-glycosylase enzymes and AP nuclease activities are necessary for YCS. a** Scheme of short-patch (SP) and long-patch (LP) BER pathways with yeast enzymes in lower case and mammalian enzymes in capitals based on recent reviews[3,4]. In SP-BER, the equivalence of yeast Pol4 and mammalian POLβ is unclear (see text). **b** Loss of *N*-glycosylase activity confers resistance to YCS. Exponentially growing wild-type (GA-8369) and *Ogg1-deg ntg1 ntg2* (GA-8457, NGΔ) cells were treated with 0.5 mM indole-acetic acid (IAA) for 4 h to provoke degradation of Ogg1 (Supplementary Fig. 3a shows proof of degradation). Cells were then treated with DMSO alone, and either NVS-BHS867[24], Zeocin, or both drugs, as indicated, for 70 min. Genomic DNA was analyzed on a CHEF gel, visualized using SYBR safe, and chromosome integrity was quantified as in Fig. 1a and summarized in Supplementary Data 1. **c** Loss of AP endonuclease activity renders cells resistant to YCS. Exponentially growing wild-type (GA-8369) and *apn1 APN2-deg RAD1-deg* (GA-8509, APΔ) cells were treated with 0.5 mM IAA for 1 h to deplete Apn2 and Rad1. Cells were then incubated with 0.5 μM CMB4563 in the presence of Zeocin as indicated for 70 min and analyzed as in (**b**).

endonuclease activity, while the Rad1-Rad10 endonuclease compensates for the absence of Apn1/Apn2 to support both LP- and SP-BER (Fig. 2a)[53]. We again made use of degron tags, creating an *apn1Δ APN2-deg RAD1-deg* strain (APΔ) and could monitor the rapid elimination of Rad1 upon the addition of IAA (Supplementary Fig. 3a). Wild-type and APΔ cells were incubated first with 0.5 mM IAA and then with Zeocin with or without the TORC2 inhibitor, CMB. The APΔ triple knock-out cells were as resistant to YCS as the NGΔ cells (Fig. 2c), suggesting that in a randomly growing culture, TORC2-induced YCS depends almost equally on AP endonucleases and *N*-glycosylases for converting nicks and base damage to DSBs.

We tested the *apn1 apn2* double mutant alone and found a weaker resistance to YCS (<2-fold) than the APΔ triple knockout, confirming that Rad1-Rad10 also contributes to the toxic processing event (Supplementary Fig. 3c). Importantly, however, we could restore YCS in APΔ cells on IAA by expressing a degron-free *APN1* from a single-copy plasmid (pAPN1; B/A = 3.8 for wild-type plus pAPN1; vs 2.8, APΔ plus pAPN1; Supplementary Fig. 3d). This suggests that Apn1 alone can drive a certain level of DSB induction, and rules out the possibility that a mutation might have arisen in the APΔ background to suppress the shattering phenotype.

Given that APE1/Apn1 are highly regulated enzymes[54] with both mitochondrial and nuclear functions, it seemed possible that TORC2-inhibition triggered YCS by increasing the level of nuclear Apn1, driving an inappropriate processing of Zeocin-induced lesions. However, the overexpression of Apn1 with an NLS, or of Apn2, or both, did not induce YCS in wild-type cells treated with Zeocin only, making it unlikely that TORC2 inhibition triggers YCS by driving nuclear uptake of AP enzymes (Supplementary Fig. 3e). Indeed, extra copies of *APN1/APN2* could not replace the BHS/CMB effect, and had weak and variable effects on YCS efficiency in exponentially growing cells (Zeocin + CMB; Supplementary Fig 3e). Thus, although loss of AP endonuclease confers resistance to YCS, elevated levels of nuclear APN1/APN2 do not drive Zeocin-induced breakage on its own, nor is cellular Apn1 concentration rate-limiting for YCS.

### Processive DNA polymerases and Trf4 drive YCS, while Pol4 and Rev3 do not

While our results clearly implicate the processing of Zeocin-induced lesions in YCS, it was unclear at which point TORC2 inhibition intervenes in the process, as actin perturbation could either enhance the early BER steps or alter downstream DNA polymerase action, interfering in repair. AP endonucleases produce an SSB with a 3′-OH that allows DNA polymerases to fill the gap in either SP- or LP-BER[3] (Fig. 2a). However, a 5-deoxyribose phosphate (5-dRP) often blocks the 5′ end, and it must be removed to complete repair. In mammalian cells, the 5′dRP lyase activity of Pol β is the predominant lyase activity for oxidized base repair by SP-BER[55]. In contrast, BER in *S. cerevisiae* appears to be primarily carried out by LP-BER, which uses the replicative DNA Pol δ and Pol ε and DNA ligase 1[56]. DNA Pol δ (Pol3) not only extends Okazaki fragments during lagging strand synthesis in yeast but is known to fill in short single-stranded gaps in various repair pathways, while its non-essential, PCNA-binding regulatory subunit, Pol32, contributes to break-induced replication[57] and AP site bypass as well[58]. Budding yeast also harbors two DNA Pol β-like enzymes, Pol4 and Trf4, with only Trf4 containing Pol β-like 5′dRP lyase activity. Yeast deficient for *pol4* are not sensitive to MMS (unlike human Pol β mutants) and instead have inefficient NHEJ[59], while *trf4Δ* yeast are hypersensitive to both MMS and $H_2O_2$[11].

Because it was unclear which of these various polymerases might be involved in YCS, we checked whether their loss conferred resistance or hypersensitivity to Zeocin in the presence of TORC2 inhibition. We note that the gene encoding the catalytic subunit of DNA Pol δ, *POL3*, is essential under unchallenged conditions, while its processivity co-factor *POL32* is not. We, therefore, tested strains lacking Pol4 (Pol β-like) and/or Pol32 for YCS efficiency. While the *pol32Δ* strain showed strongly reduced fragmentation (B/A = 1.9 vs 6.2 in wild-type; Fig. 3a), *pol4Δ* behaved like wild-type (B/A = 8.3; Fig. 3a). The *pol4Δ pol32Δ* double mutant showed roughly the same resistance to YCS as *pol32Δ* (B/A 1.3 vs 1.9; Fig. 3a). Thus, consistent with a role in LP-BER, Pol δ contributes to YCS, i.e., the conversion of oxidative lesions to DSBs, in cycling cells.

Pol32 increases the stability of the Pol3 and Pol31 complex, thereby promoting DNA Pol δ processivity[38,60]. A short C-terminal deletion in Pol3 also compromises Pol δ processivity by interfering with Pol3 interaction with Pol31 (*pol3-ct*)[61]. We, therefore, tested two independent *pol3-ct* isolates for their impact on YCS. Importantly, both showed resistance to YCS at low Zeocin levels (Fig. 3b), although, as expected, at higher levels of Zeocin fragmentation could still occur. This argues that Pol δ processivity may contribute to YCS. We checked the role of another Pol32-binding polymerase, the trans-lesion synthesis polymerase Pol ζ or Rev3[62], but *rev3* deletion did not alter YCS efficiency (Fig. 3c).

It has been proposed that DNA Pol ε, encoded by the essential gene *POL2*, functions redundantly or alongside Pol δ in BER[9]. We, therefore, tested the temperature-sensitive *pol2-18* allele, which conditionally inactivates the Pol ε catalytic activity at 37 °C in the YCS assay. The *pol2-18* mutation conferred a very minor resistance to YCS, less pronounced than that of *pol32Δ* (Fig. 3d), suggesting that Pol ε and δ may act redundantly in this repair function. Unfortunately, yeast is not viable if mutated for both DNA polymerases. Finally, we tested the Pol β-like polymerase Trf4, which is thought to process the blocked 5′ end with its 5′dRP lyase activity following strand cleavage by AP endonuclease, in a process called "gap tailoring"[11]. Intriguingly, in the absence of Trf4, we found some fragmentation upon Zeocin treatment even without TORC2 inhibition (arrow, Fig. 3e), suggesting a unique role for Trf4's 5-dRP lyase activity in BER, possibly as a prerequisite for re-ligation. Nonetheless, *trf4Δ* cells are partially resistant to the YCS triggered by Zeocin and CMB (B/A 0 4.0 vs 10.0 in wild-type, Fig. 3e). This reduction may reflect a second activity attributed to Trf4: that of loading the more processive DNA Pol ε[63]. If Trf4 were either to load DNA pol ε or rapidly extend the 3′OH end, it could convert a ss nick into a DSB by elongating until it encountered a nick on the other strand. Its loss would thus reduce polymerase processivity in LP-BER.

### The incorporation of oxidized nucleotides is not involved in YCS

A further pathway relevant to cellular sensitivity to oxidation is the incorporation of free oxidized nucleotides during DNA synthesis, a pathway that has been reported to generate genome instability and cytotoxicity[64,65]. Loss of hMTH1, which hydrolyzes oxidized purine dNTPs in the nucleotide pool and thus prevents their incorporation into DNA[66], sensitizes cells to oxidation in a manner partly dependent on Pol β[65], suggesting that the incorporation of oxidized free nucleotides (e.g., 8-oxo-G) drives DNA damage. We tested this in the context of YCS by eliminating the yeast 8-oxo-dGTP diphosphatase (PCD1), the homolog of hMTH1, generating a larger pool of free oxidized purine nucleotides. This, however, did not alter YCS rates (Supplementary Fig. 4). We conclude that the conversion of Zeocin-induced damage into breaks reflects impaired long-patch BER, with contributions from glycosylase activity, endonucleolytic attack and the processivity of DNA polymerases.

### YCS in G1-phase cells depends on AP endonucleases, but not *N*-glycosylases

In Fig. 1b we showed that YCS could occur both in G1-phase arrested and in cycling cells[24], but it remained unclear whether the factors that drove YCS were the same throughout the cell cycle. To test this, exponentially growing wild-type, APΔ and NGΔ cells were arrested in G1 and incubated with Zeocin, CMB, or both (Fig. 4a, b). Surprisingly,

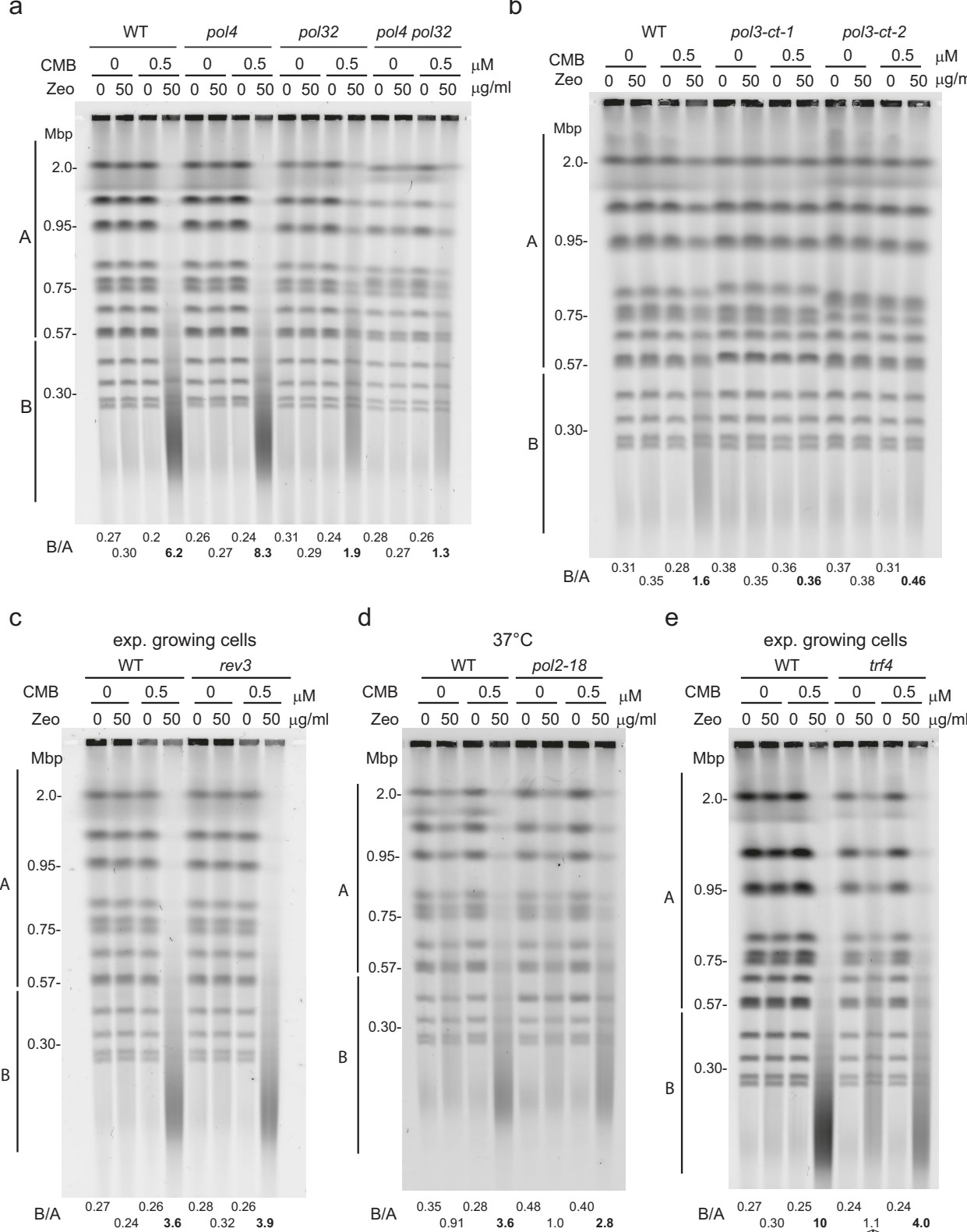

whereas the loss of AP endonuclease activities conferred resistance in G1 cells as well as in cycling cells (Fig. 4b), the depletion of *N*-glyco-sylase activity had no effect in G1 (Fig. 4a). To ensure that there were no spurious mutations arising in NGΔ, we split one culture of NGΔ cells, and either arrested them in G1 phase or left them unsynchronized prior to the YCS assay. Again, the NGΔ cells were not resistant to YCS in the G1 phase, although they were in the exponential culture

(Fig. 2b). Ruling out artefacts arising from reduced Zeocin uptake or function in one phase or another, we found that the number of AP sites in both G1-phase and cycling cells was roughly equal (Supplementary Fig. 2a).

The fact that loss of *N*-glycosylases does not render cells in G1 resistant to YCS is intriguing, as G1-phase cells are more sensitive to Zeocin than randomly growing cells, at least when tested in a damage

**Fig. 3 | Impaired processivity of DNA Pol δ and Pol ε but not Pol4 or Rev3, reduces YCS efficiency. a** Strains lacking DNA Pol δ subunit Pol32 are partially resistant to YCS. Wild-type (GA-1981), *pol4Δ* (GA-10595), *pol32Δ* (GA-9686), and *pol4Δ pol32Δ* (GA-10697) mutants were exponentially cultured in SC medium. Cells were treated with 0.5 μM CMB4563, 50 μg/ml Zeocin, or a combination of both reagents for 60 min. Chromosomal integrity was analyzed and quantified as in Fig. 1a. **b** As 3a, except wild-type (WT, GA-1981), and two independent isolates of pol3-ct (*pol3-ct-1* and *-2*; GA-10997 and GA-11000) were exponentially grown in SC medium, and were treated with 0, 50, or 100 μg/ml Zeocin with or without 0.5 μM CMB4563 as indicated for 80 min. Chromosomal integrity was analyzed and quantified as in Fig. 1a and in Supplementary Data 1. **c** As 3a, except that isogenic wild-type (WT, GA-1981) and *rev3Δ* (GA-9683) cells were grown exponentially and incubated with the indicated reagents for 80 min, prior to CHEF gel analysis and

quantitation as in Fig. 1a and in Supplementary Data 1. **d** Reduced DNA Pol ε activity confers partial resistance to YCS. Wild-type (GA-741) and *pol2-18* (GA-742) cells (see[96]) were exponentially cultured in SC at 25 °C. Cell culture was shifted to 37 °C for 40 min, treated with 1 μM CMB4563, 75 μg/ml Zeocin, or a combination of both reagents for 70 min, prior to CHEF gel analysis and quantitation as in Fig. 1a and in Supplementary Data 1. Partial resistance to chromosome shattering is seen under YCS conditions. **e** Loss of Trf4 enhances Zeocin sensitivity but reduces YCS. Wild-type (GA-1981) and *trf4Δ* (GA-10632) cells were exponentially cultured in SC, then treated with 50 μg/ml Zeocin with or without 0.5 μM CMB4563, as above for 80 min, prior to CHEF gel analysis and quantitation. In the absence of CMB, *trf4Δ* shows increased chromosome fragmentation on Zeocin (B/A = 1.1 vs. 0.3 in WT), but under YCS conditions, *trf4Δ* is partially resistant (B/A = 4.0 vs. 10 for WT). CHEF gel quantitation as in Fig. 1a and Supplementary Data 1.

survival assay (Fig. 4c, d). Although neither random nor arrested cultures lost viability upon transient TORC2 inhibition, the G1-phase cells were very sensitive to Zeocin even when TORC2 was active. Underscoring these cell cycle differences, strains lacking all endonucleases (APΔ) showed improved colony survival after transient Zeocin and CMB treatment of G1-phase cells, while in exponential cultures, the NGΔ strain survived better than wild-type upon Zeocin and CMB exposure (Fig. 4c, d). Put simply, it appears that AP endonucleases are the primary cause of DSBs in G1, while both *N*-glycosylases and AP endonucleases contribute to cycling cells, leading to irreversible cell death under YCS conditions. The fact that NGΔ strains still show robust YCS in G1, argues that glycosylases are unlikely to be the crucial target of TORC2 inhibition, whereas AP endonucleases, whose ablation rendered cells resistant to YCS in both G1-arrested and exponential cultures, could be. Not surprisingly, the loss of all AP endonucleases does not increase general Zeocin survival, even though it reduces YCS (Fig. 4d vs Fig. 2c).

A previous report had implicated Apn1 in an Ogg1-independent repair of 8-oxo-G residues thanks to its processive 3′ to 5′ exonuclease activity[67]. This was defined as a nucleotide incision repair (NIR) pathway, and it showed strong dependence on Apn1's nuclease activity, which processes modified bases to generate a 3′ OH end compatible with DNA polymerization[15]. Interestingly, elevated levels of NLS-Apn1 and NLS-Apn2 did provoke higher Zeocin sensitivity in yeast by drop assay, but only slightly increased YCS in G1 cells (Supplementary Fig. 6a,b; B/A = 4.7 vs 3.1 upon elevated levels of nuclear Apn1/Apn2). Thus, consistent with Fig. 4, high-level AP endonuclease activity aggravates survival on Zeocin and weakly augments YCS. This may be through NIR. As mentioned earlier, elevated levels of nuclear Apn1 or Apn2 alone did not increase chromosome breakage on Zeocin, arguing that TORC2 might upregulate AP endonuclease activity in ways other than enzyme abundance and localization (Supplementary Fig. 6b).

### DNA Pol δ subunit Pol32 is not essential for YCS in G1- arrested cultures

We next examined the roles of the different DNA polymerases in YCS in G1-arrested cells. Unlike the situation in cycling cells, neither *pol32Δ* nor the *pol4Δ pol32Δ* double mutant conferred resistance to YCS in G1-phase cells (*cf.* Figs. 5a, 3a), while the *trf4Δ* strain showed significant resistance to YCS in G1-phase cells (Fig. 5b, B/A = 3.8 *trf4Δ* vs 11.0 wild-type), as observed in cycling cells (Fig. 3e). The impact of Pol ε loss in the ts mutant *pol2-18* was only minor (Fig. 5c). The fact that Pol32 influences YCS in exponential cultures, but not during G1 arrest, could either indicate that DNA Pol δ is less important than the Trf4-Pol ε axis in G1 phase, or else that in S phase the relative abundance or availability of the DNA polymerases changes. We suggest that in G1 phase Trf4 replaces Pol32-Fen1/Rad27, which is thought to act in S phase[19] to remove 5′RP, increasing the potential for mis-coordinated repair of neighboring lesions.

In yeast, both SP- and LP-BER require the action of yeast ligase 1 (Lig1). It is expected that the ablation of *CDC9*, which encodes this essential ligase, would enhance breakage rather than render chromosomes resistant to YCS since Lig1 is needed for the last step in BER. Using *cdc9-1*, a conditional allele that inactivates the ligase at 37 °C, we indeed saw strongly enhanced YCS at 37 °C, even in G1-arrested cells (Fig. 5d). As expected, some chromosome fragmentation occurs in both wild-type and *cdc9-1* cells even without CMB (Fig. 5d). However, when Zeocin and TORC2 inhibitor were combined fragmentation was even more extensive without Lig1 activity, and there was a reduction in average fragment size to ~20 kb, rather than 100–120 kb (Fig. 5d; corresponding to roughly 600 DSBs per genome, Supplementary Fig. 2c). This argues that other pathways of repair that depend on Lig1 (NHEJ, HR, BIR, NER) are functioning under YCS conditions, resulting in even more unrepaired damage when *cdc9-1* is inactivated. Given that Lig1 is largely active in the presence of TORC2 inhibitors, we can conclude that Cdc9/Lig1 is itself not the target of TORC2 regulation. Indeed, we find Cdc9/Lig1 hyperphosphorylated under YCS conditions[46], possibly reflecting its upregulation.

### The loss of end-processing confers variable results on YCS resistance in the G1 phase

The final set of repair factors that might be involved in YCS are those that clean up nonactionable DNA nicks to enable Lig1 to act. As stated above, in mammalian cells, Pol β can remove 2-deoxyribose-5-phosphate or 5′-AMP-dRP from the 5′ termini of BER intermediates[55], while in yeast, Trf4 mediates this event. Cleavage of a flap by Fen1/Rad27 also produces ends that can be ligated. To test the importance of other enzymes that can "clean" processed damage in YCS, we tested deletions of *TPP1*, which encodes a protein that removes 3′ phosphates at strand breaks, *HNT3*, which encodes aprataxin, an enzyme that reverses DNA adenylation (5′-AMP-dRP), and *TDP1*, which encodes an enzyme that hydrolyzes 3′ and 5′ phosphotyrosyl bonds, such as those created by cleavage by topoisomerases[68–70]. The loss of Tpp1 led to enhanced chromosome breakage, like the loss of Lig1, confirming the importance of 5′ end processing of Zeocin-induced damage for the final step of repair (Supplementary Fig. 6c). The loss of Hnt3, which acts on the 5′AMP-dRP similar to Trf4, led to slight resistance under YCS conditions like Trf4, and the loss of Tdp1 had a very minor, likely negligible, impact (Supplementary Fig. 6d). Only *tpp1Δ* had an effect in the absence of TORC2 inhibitors, slightly increasing fragmentation in G1 phase cells, like *cdc9-1* (Supplementary Fig. 6c). We conclude that 5′ end processing impacts YCS weakly: Trf4 and Hnt3 enhance break accumulation, and Tpp1 attenuates it slightly. Although the mechanism is unknown, it is possible that these enzymes help regulate the processing of a second adjacent lesion until the first is successfully ligated, to prevent the coincident repair of clustered lesions.

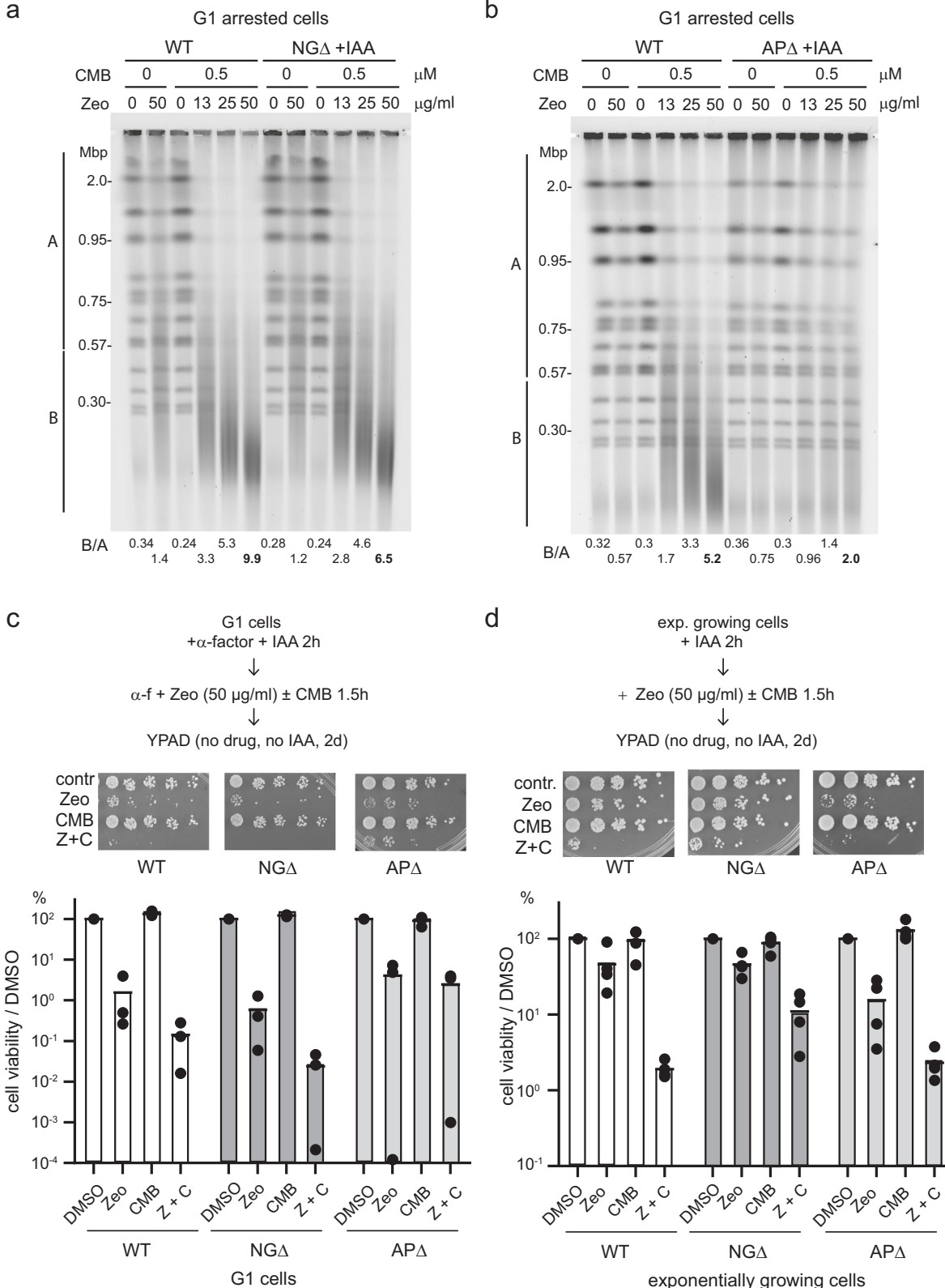

## Nuclear actin modulates Zeocin sensitivity

Three potential models could explain the massive generation of DSBs from Zeocin-induced oxidation in the presence of TORC2 inhibition and/or actin depolymerization. First, Apn1/Apn2 activities may be aberrantly upregulated, such that a premature processing of an adjacent lesion prior to the fruitful ligation of the first, allows DSB formation. This, however, does not seem to require or even reflect elevated levels of Apn1 or Apn2 in the nucleus but may reflect increased

accessibility of damage to the nuclease. Second, YCS conditions could render DNA polymerases exceptionally processive, driving the first repair event into the nick on the other strand prematurely. This processivity could arise from intrinsic polymerase misregulation, or from extrinsic activities, such as nucleosome remodeling. It is well established that nucleosome remodelers regulate fork progression, both in normal replication and repair-coupled elongation[39,40]. The remodelers involved in such activities (e.g., INO80C and BRG1) intriguingly contain

**Fig. 4 | AP endonucleases, but not N-glycosylases, are required for YCS in G1 cells. a** N-glycosylase activity is largely dispensable for YCS in G1-arrested cells. Exponentially growing wild-type (GA-8369) and *Ogg1-deg ntg1 ntg2* (NGΔ; GA-8457) cells were treated with 0.5 mM IAA for 2 h (Ogg1 degradation in Supplementary Fig. 3a), then α-factor was added for 2 h to the arrest the culture in G1 (FACS analysis in Supplementary Fig. 5). Cells were then treated with DMSO or CMB4563 in the presence of the indicated amounts of Zeocin for 80 min, prior to CHEF gel analysis and quantitation as in Fig. 1a. **b** YCS occurs in an AP-endonuclease-dependent manner in G1. Exponentially growing wild-type (GA-8369) and *apn1 APN2-deg RAD1-deg* (APΔ; GA-8509) cells were treated with 0.5 mM IAA and α-factor for 2 h (FACS analysis in Supplementary Fig. 5; Rad1 degradation control in Supplementary Fig. 3a). Cells were treated with CMB4563 and Zeocin as indicated for 80 min, prior to CHEF gel analysis and quantitation as in Fig. 1a and Supplementary Data 1. **c** Differential sensitivity of G1 cells and exponential cultures to Zeocin. Wild-type

(WT), NGΔ, and APΔ cells were grown in complete minimal media (SC) and arrested in G1 by α-factor together with 0.5 mM IAA for 2 h (FACS analysis in Supplementary Fig. 5). Cells were then treated with 50 μg/ml Zeocin, 0.5 μM CMB4563, or both drugs for 90 min. Cells were spotted on YPAD in a 5-fold dilution series. Images taken after 2 days were quantified for cell survival percentages normalized to the DMSO control. Data points are from 3 biological replicates. **d** Wild-type, NGΔ, and APΔ cells were exponentially grown in SC and treated with 0.5 mM IAA for 2 h to deplete degron-tagged proteins, and then treated as panel (**c**). Data points are from 4 biological replicates. AP endonuclease loss increases survival of YCS conditions in G1 arrested cells, consistent with the resistance to chromosome breakage, but not in exponential cultures. N-glycosylase activities (Ogg1, Ntg1, Ntg2) loss aids survival of Zeocin and TORC2 inhibitors in exponentially growing cells, but not in G1-arrested cultures.

---

actin as an essential subunit[36]. Enhanced nucleosome remodeling could also increase access to processing enzymes, like Apn1, leading to the premature processing of a neighboring lesion. A third option is that there is impaired communication between the first and second repair events, for example, interference in the formation of a repair focus that might coordinate the processing of paired lesions. As suggested by the fact that Latrunculin A, which depolymerizes cytoplasmic actin filaments, mimics the presence of BHS/CMB on Zeocin[24], we next tested if excess nuclear actin itself interferes in BER.

Genomic mutations of actin cause cytoskeletal defects that affect every aspect of yeast metabolism and growth, including the cell cycle and bud emergence. This makes it very difficult to use genomic *act1* mutants for assessing the role of the minor population of nuclear localized actin. We, therefore, used an ectopically expressed, inducible form of actin in which both nuclear export signals are mutated, *act1^nes*[71]. This protein accumulates in the nucleus without interfering with cytosolic actin function. Accordingly, the induction of *act1^nes* exhibited only a minor growth phenotype, while expression of wild-type *ACT1* from a multicopy plasmid, led to cell growth inhibition even in the absence of Zeocin (Fig. 6a, no DOX). While unperturbed growth was normal, *act1^nes* expressing cells were highly sensitive to Zeocin (Fig. 6a). Chromatin fractionation of yeast cells[72] into nuclear pellet (Chr for chromatin) and supernatant (S for cytosol and soluble nuclear proteins), showed that neither the endogenous actin nor the overexpressed Act1 was chromatin-associated, while the overexpressed, act1^nes protein was strongly associated with the chromatin pellet (Fig. 6b). This is consistent with its expected nuclear accumulation and its likely integration into chromatin-modifying complexes.

We further mutated the plasmids encoding DOX-repressed *act1^nes* in order to test whether actin polymerization capacity plays a major role in Zeocin sensitivity. We generated the filament-favoring allele *act1-S14C^nes*[73], and a polymerization-deficient allele *act1-AP^nes*, which carries two mutations (A204E/P243K) at the pointed end[74]. All were expressed to equivalent levels from a single-copy plasmid upon removal of doxycycline (no DOX, Fig. 6c). Intriguingly, expression of all *act1^nes* genes strongly activated the Rad53 checkpoint on Zeocin, but not in its absence, mimicking the elevated checkpoint response observed when TORC2 inhibitors are combined with Zeocin (Fig. 6d). This argues that nuclear actin accumulation per se does not induce damage, yet like TORC2 inhibition, nuclear actin appears to block the repair of Zeocin-induced lesions, elevating the DNA damage checkpoint response (Fig. 6d). We note that the polymerization-deficient form of nuclear actin (AP^nes) had slightly weaker impact than the polymerization competent forms.

To see if *act1^nes*, *act1-S14C^nes*, or *act1-AP^nes* expression alters chromosome fragmentation in the presence of Zeocin, we performed quantitative CHEF on the appropriately treated strains with and without TORC2 inhibition. Remarkably, the nuclear enrichment of all forms of actin, including a further allele *act1-111^nes*, led to chromosomal breakage in the presence of Zeocin, even without CMB

(Fig. 6e). Nonetheless, TORC2 inhibition enhanced the effect, especially for *act1-AP^nes* and *act1-111^nes* (Fig. 6e). This shows that actin accumulation in the nucleus per se – both wild-type and mutant – helps convert Zeocin-induced lesions to DSBs. While it may not be the only relevant repercussion of TORC2 inhibition, this result suggests that nuclear actin plays a fairly direct role in the YCS-driven BER pathology.

### Actin interaction with BER enzymes and INO80C

We next asked if actin directly binds BER factors, in particular AP enzymes or the DNA polymerases that drive fragmentation under YCS conditions. Epitope tags on actin lead to partial inactivation, therefore we made use of novel actin specific bicyclic peptides isolated from a phage display library, that bind either F-actin (A18) or G- and F-actin (A15)[75]. These bicyclic peptides are circularized sequences of 10–17 amino acids with highly selective binding specificity validated both in vitro and in vivo[75]. Magnetic bead-coupled bicyclic peptides A18 and A15, as well as the TATA2 control, were used to recover ligands from a total yeast cell extract (see "Methods"). After rigorous washing, the recovered proteins were visualized by silver staining, and the most prominent peptides were identified by mass spectrometry (Fig. 7a). We found that A18 and A15 both efficiently bound actin and pulled down the abundant cytoplasmic actin-binding proteins from the cell extract (e.g., Pan1, Sla2, Las17, and Cofilin). Interestingly, only A15 (which binds both F- and G-actin) efficiently bound two subunits of the ssDNA binding factor, RFA. Cofilin was recovered with both peptides, albeit preferentially with A15. Using western blots to probe the bicyclic peptide-recovered factors, we detected Apn1 in both the A18 and A15 pull-down fractions, but not Mcm2 or tubulin, while Ogg1 was only detected in the A15 pull-down (Fig. 7a).

Given the potential Apn1-actin interaction, we next asked whether Apn1 was increased either in nuclei or in a chromatin-bound fraction under YCS conditions. Using a functional GFP-fusion to Apn1, we scored its nuclear intensity comparing normal growth and YCS conditions using live imaging. Rather than detecting an increase, we detected a slight decrease in nuclear Apn1 (Fig. 7b). We reasoned that despite reduced nuclear levels of Apn1, its binding to chromatin might be enhanced. Therefore, we fractionated yeast into chromatin-bound and soluble protein under YCS conditions[72]. Again, chromatin-bound Apn1 levels were relatively low and did not change during YCS (Fig. 7c). In parallel, however, we probed a core subunit of INO80 complex, Arp5, a component of the essential nucleosome-binding domain of INO80C, which contributes to the processivity of DNA polymerases after recovery from replication fork stalling[39–42]. In contrast to Apn1, we saw enhanced levels of Arp5 on chromatin under YCS conditions (normalized to Orc2; Fig. 7c, d).

Arp5 is an actin-fold-containing protein, which, like actin, is an integral component of INO80C[36]. Actin itself is found as an Arp8-Arp4-actin module that binds the helicase/SANT-associated domain of Ino80 as well as DNA[28,76,77]. The actin-related proteins and actin

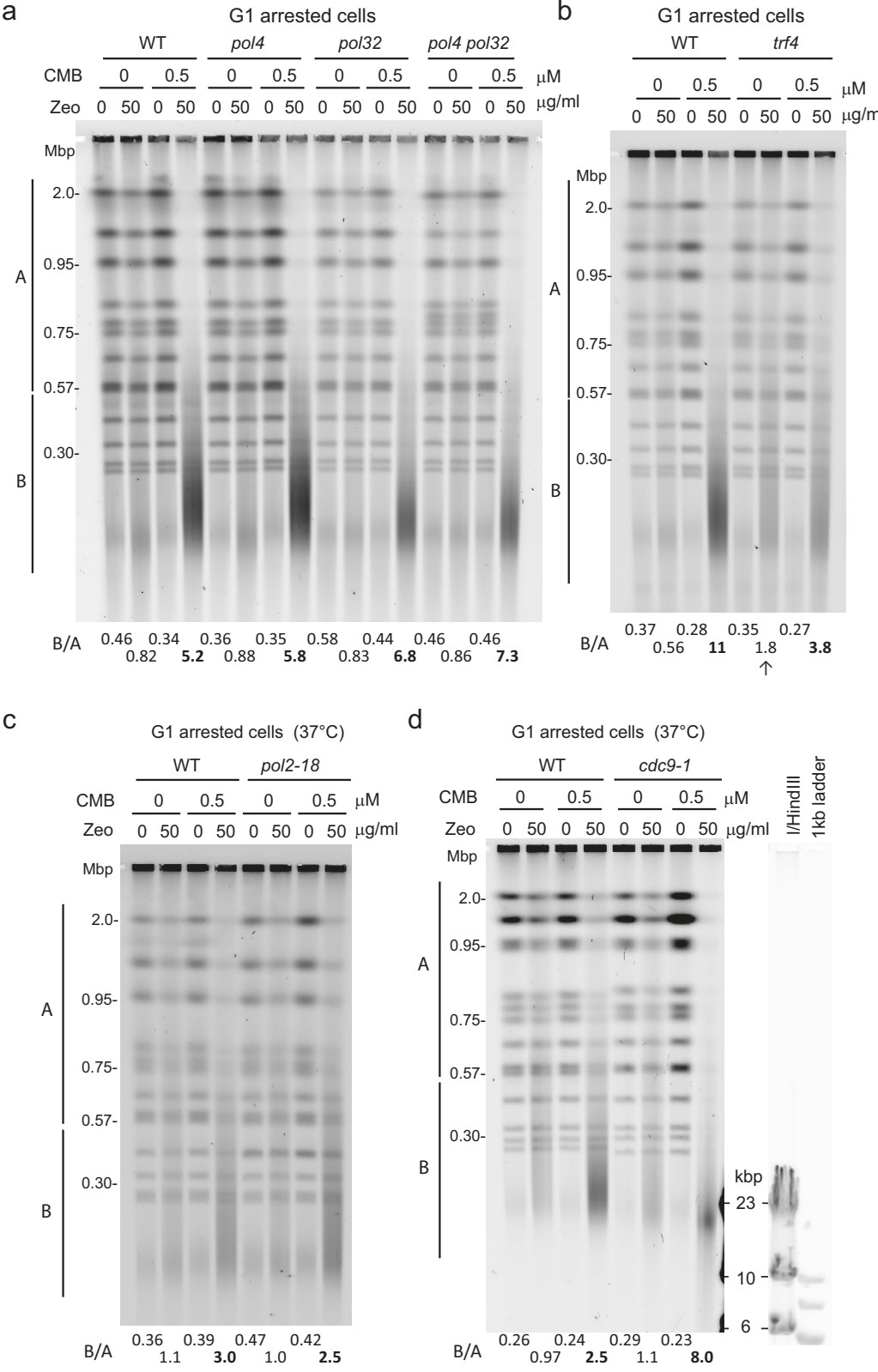

itself are essential for nucleosome eviction and degradation in response to high dose Zeocin[50] and support DNA polymerase processivity[39–42]. We therefore checked whether chromatin is more open, i.e., more accessible, under YCS conditions by using low-level expression of a bacterial DAM methylase. DAM methylates adenines in the GATC context[78], and if DNA is protected by either nucleosomes or non-histone factors, one detects a decrease in $^mA$, by

monitoring sensitivity to restriction enzyme *DpnI* cleavage (Fig. 7e). In five independent repeats, we detected a reproducible increase in DNA accessibility on CMB or CMB plus Zeocin, but not upon the addition of low-level Zeocin alone (Fig. 7e). This suggests that reduced levels of cytoplasmic F-actin can trigger a more open chromatin state, which has repercussions on step-wise repair at clustered lesions.

**Fig. 5 | Loss of Trf4 and DNA Pol ε reduce YCS in G1, while Lig1 loss increases it. a** Ablation of *POL32* does not confer YCS resistance in G1-arrested cells. Wild-type (WT, GA-1981), *pol4Δ* (GA-10595), *pol32Δ* (GA-9686), and *pol4Δ pol32Δ* (GA-10697) mutants were arrested in G1 by α-factor for 100 min (Supplementary Fig. 5 for FACS). Cells were then treated with 0.5 μM CMB4563, 50 μg/ml Zeocin, or both for 70 min prior to CHEF gel analysis and quantitation as in Fig. 1a; and Supplementary Data 1. **b** Wild-type (WT; GA-1981) and *trf4Δ* (GA-10632) cells were exponentially cultured in SC, then arrested in G1 with α-factor. G1-arrested cells were treated checked, and analyzed by CHEF gel and quantitation as in 5a. **c** Isogenic wild-type (WT; GA-741) and *pol2-18* (GA-742) cells[96] were exponentially cultured in SC at 25 °C, then arrested

with α-factor. Cultures were shifted to 37 °C for 45 min, then treated with Zeocin CMB4563, or both, for 70 min prior to CHEF gel analysis and quantitation as above. **d** Loss of DNA ligase I (Cdc9) activity greatly enhances YCS. Isogenic wild-type (WT, GA-8709) and *cdc9-1* (GA-8708) cells were grown exponentially in SC at 25 °C. Cells were arrested in G1 by α-factor for 2 h, then shifted to 37 °C for 45 min. Cells were treated with Zeocin, CMB4563, or both, as indicated for 60 min prior to CHEF gel analysis and quantitation as above. FACS analysis in Supplemental Fig. 5. Markers shown were run alongside the samples to determine the size of fragments produced in the *cdc9-1* sample.

## Discussion

We have reported a rapid and irreversible fragmentation of the yeast genome following the exposure of wild-type cells to low doses of Zeocin and non-toxic levels of a TORC2 inhibitor, whereas neither treatment alone compromised genomic integrity[24]. Hurst et al. show that G-actin levels in the nucleus increase in the presence of TORC2 inhibitors or upon deletion of the cytoplasmic actin chaperone Las17[46]. Consistently, we find that elevated levels of nuclear actin, either as a form that favors or disfavors polymerization, is sufficient to trigger YCS in the presence of Zeocin without TORC2 inhibition (Fig. 6e). By systematically testing repair mutants, we found that the generation of these irreparable DSBs requires processing of the oxidized base by N-glycosylases and/or AP endonucleases, which convert oxidized bases to single-strand nicks (Fig. 2). Downstream of AP endonuclease action, DNA polymerase processivity contributes to the generation of DSBs, suggesting that breaks arise from miscoordinated LP-BER of clustered lesions (Fig. 3). Religation by Lig1 (Cdc9) after DNA synthesis is essential for restoring genomic integrity, but inactivation of Cdc9 leads to hyperfragmentation during YCS, and not resistance, as observed with the other repair factor mutants (Fig. 5d). Whereas actin is shown to bind Apn1, it is well established that nuclear actin is primarily found in nucleosome remodeler complexes, such as INO80C, which promote DNA polymerase processivity[39–42]. We propose a model in Fig. 8 that unites these observations and explains how the uncoordinated repair of adjacent lesions can generate DSBs.

Earlier work postulated that paired or clustered lesions must be repaired in a stepwise manner to avoid the generation of irreparable breaks[19,20,79] (reviewed in ref. 18). Furthermore, DSBs were found to occur in plasmids treated with a DNA methylating agent, N-methyl-N-nitrosourea (MNU) following repair in *Xenopus* egg extracts, due to the coincident action of mismatch repair and BER[80]. This linearization, like the phenomenon we describe here, occurs without heat depurination and alkali. Our work argues that coordinated and delayed processing of clustered base oxidation events is a conserved phenomenon, that occurs not only on plasmids but on genomic DNA. This process is particularly relevant to damage induced by bleomycin-related antibiotics like Zeocin, which induce closely positioned oxidative damage on opposite strands[25]. Indeed, we show that at high concentrations Zeocin itself can drive the linearization of a plasmid, albeit inefficiently, and that the action of the human glycosylase hOGG1 strongly increases DSB generation in vitro following treatment with low Zeocin concentrations (Fig. 1e). Thus, while the need for sequential, step-wise repair is not restricted to Zeocin-induced damage, the clustering of lesions, like those induced by Zeocin, renders the timing or coordination of repair events crucial for genome integrity.

We have systematically explored the mechanism(s) that coordinate repair events to prevent the conversion of Zeocin-induced base oxidation into breaks, by deleting factors involved in lesion processing[3]. The AP endonucleases, encoded by *APN1* and *APN2* in yeast, are key players in the generation of these breaks. It is well known that Apn1/Apn2 and the

human APE1 enzymes have complex and highly regulated roles in the DNA damage response[15,81]. Mammalian APE1 enzyme not only cleaves abasic sites generated by glycosylase-mediated excision of damaged bases but, together with Thymine DNA Glycosylase (TDG), catalyzes the first steps in the demethylation of cytosines[82]. In this context, APE1 triggers the release of TDG and other glycosylases from abasic sites. However, the release of glycosylases does not appear to be the main role of Apn1/Apn2 in YCS, given that NGΔ cells were not resistant to YCS in the G1 phase, while APΔ cells were (Fig. 4).

Both mammalian and yeast AP endonucleases also participate in the BER/NIR switch[15,83,84], whereby APE1 (or Apn1/Apn2) bypasses the action of the glycosylase and generates a nick that retains the 5′ damaged nucleotide and generates a 3′OH group. In this function, AP enzymes can recognize and process a diverse set of lesions, including oxidized pyrimidines, formamido-pyrimidines, exocyclic DNA bases and uracil, bulky lesions, and UV-induced 6–4 photoproducts[82]. The switch from a glycosylase-dependent to a glycosylase-independent cleavage mode appears to reflect allosteric changes in APE1 structure[83]. This may occur in the G1 phase in yeast and would explain why NGΔ cells are not resistant to YCS conditions, while APΔ cells are. We also note that yeast Apn1 and Apn2 possess 3′ to 5′ exonuclease activity[67], which allows conversion of ss lesions into DSBs if the AP endonuclease extends the gap on one strand until it encounters a nick at a nearby lesion on the opposite strand (Fig. 8). In this sense, the eviction or attenuation of Apn1/Apn2 activity may be important to prevent the conversion of ss-lesions into DSBs.

Although we do not see elevated levels of nuclear Apn1/Apn2 under YCS conditions, Apn1 binds actin in yeast cell extracts (Fig. 7a). Intriguingly, after exposure to Zeocin or Zeocin and LatB, human APE1 shows slightly reduced nuclear abundance[38], and the same was observed here for yeast Apn1 (Fig. 7b–d). Moreover, overexpression of NLS-Apn1 does not mimic TORC2 inhibition, leading to a very minor increase in DSBs under YCS conditions (Supplementary Figs. 3, 6). Rather than increasing nuclear AP endonuclease abundance, we propose that the loss of cytoplasmic actin filaments (accompanied by an increase in nuclear actin) promotes lesion access for the AP endonucleases by evicting or shifting nucleosomes through actin-dependent remodelers (Fig. 8).

As mentioned above, likely targets of this pathway are DNA polymerases and chromatin remodelers. In *Xenopus* extracts, DNA Pol δ was recovered as an actin binder, together with a range of chromatin modifiers or remodeling enzymes[85] (namely, TRRAP, RuvB1, BRG1, BAF155). We, however, did not detect DNA pol δ in our actin-pulldown assay (Fig. 7a). Nucleosome remodelers are known to augment the processivity of DNA polymerases by removing or evicting nucleosomes in front of moving forks[41], and yeast INO80C specifically promotes replication polymerase processivity at both normal and collapsed forks[39–42]. Given that actin is an integral component of INO80C[28,76,77], and the actin-Arp4 complex may be rate-limiting for its activity[86], we favor the notion that enhanced polymerase activity and premature processing of the second lesion arises from elevated

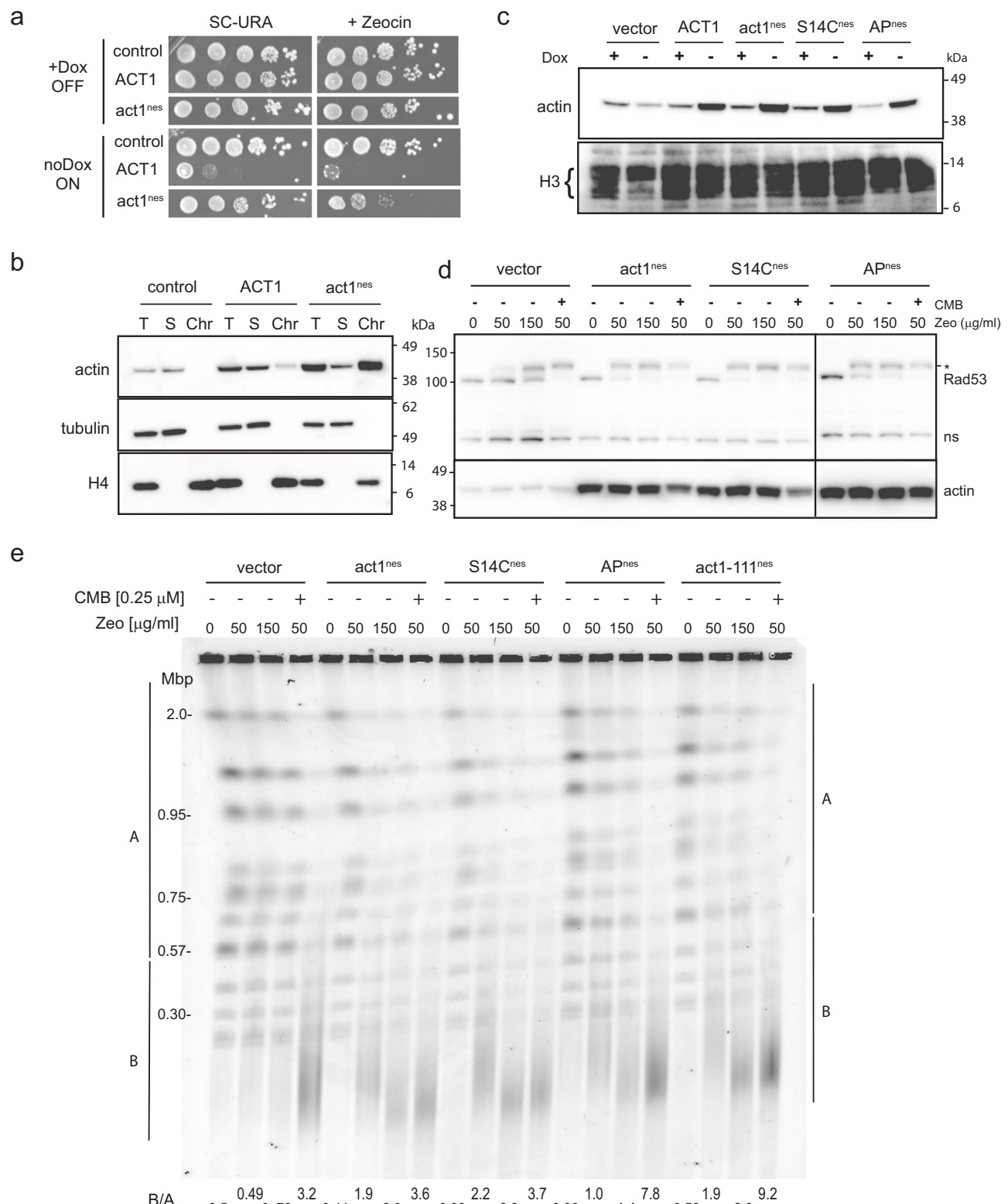

activity of INO80C, which is influenced by a nuclear influx of G-actin. Upon TORC2 inhibition, this is triggered by the depolymerization of cytoplasmic actin filaments[46].

Consistent with this model, we find that reducing DNA Pol δ processivity (by deleting Pol32 or in the *pol3*-processivity mutant *pol3-ct*) reduces the conversion of Zeocin-induced lesions to DSBs (Fig. 3). Similarly, we find YCS-resistance in cells lacking Trf4, a DNA pol β-like enzyme that can cleave off the 5′dRP to allow end ligation and help load the more processive DNA Pol ε[11], for LP-BER. On the other hand, we rule out roles for the repair polymerases Rev3 and Pol4. We propose that in our current system, enhanced INO80C activity may provide better access of Apn1 to the second lesion, prematurely generating an SSB, as well as enabling DNA Pol δ and ε processivity (Fig. 8).

**Fig. 6 | Nuclear actin overexpression sensitizes cells to Zeocin, inducing DSBs.** **a** Wild-type GA-1981 cells were transformed with pCM190 vector (control), pCM190 carrying *ACT1* (wild-type), or *act1-nes*[71] (*ACT1* bearing two mutated nuclear export signals), and were selected on SC-URA + 1 µg/ml doxycycline (Dox) to suppress pCM190 plasmid expression[97]. The colonies transformed were diluted in a 5-fold series and spotted on SC-URA +/- 60 µg/ml Zeocin in the presence (+ Dox, OFF) or absence (no Dox, ON) of 1 µg/ml doxycycline. Growth was imaged after 3 days at 30 °C. **b** Chromatin fractionation shows that overexpressed act1-nes co-purifies with chromatin. Exponentially growing GA-1981 cells transformed as in (**a**) were subjected to chromatin fractionation[72]. Overexpressed WT actin (Act1) was mostly found in the chromatin unbound fraction, while the majority of act1-nes co-purified with chromatin (as histone H4), indicating that act1-nes is retained in the nucleus and may be chromatin-bound. Fractionation controls and blots are in Source data files. **c** Actin mutants act1-S14C-nes, act1-AP-nes and Act1-nes (see text) are expressed like Act1-nes. The total extracted protein was subjected to Western blots

with anti-actin and anti-H3. Original blots are provided in the Source Data file. **d** Strains expressing act1-S14C-nes, act1-AP-nes and Act1-nes hyperactivate Rad53 kinase on Zeocin vs vector control (wild-type). The total protein sample was subjected to Western blot probed with anti-Rad53 and anti-actin on the same blot (see Materials). Rad53 shows a significant shift due to phosphorylation by Mec1 and autophosphorylation[94]. ns indicates a nonspecific cross-reactive band. Full blots are included in Source Data files. **e** The same strains bearing the indicated plasmids (see **a** and **c**) were exponentially cultured in SC-URA + 1 µg/ml Dox, then washed twice with SC-URA without Dox and cultured for 6 h without Dox to induce actin. Cells were treated with 0, 50, 150 µg/ml Zeocin for 80 min, and genomic DNA was subjected to CHEF gel and stained with HDGreen. Also tested was another actin mutant, *act1-111*[98], bearing the *nes* mutation. This actin mutant is polymerization-competent but can provoke YCS in the absence of CMB[46]. Quantitation of YCS as in Fig. 1a and Supplementary Data 1.

This role for DNA Pol δ processivity contrasts with an earlier study showing that *pol32Δ* enhanced the formation of DSBs after MMS-induced alkylation in yeast[19]. These authors suggested that in the case of alkylation by MMS, the binding of Pol32 to Rad27 (FEN1 in mammals) during the first repair event might physically impair the second, by preventing the release of this flap endonuclease[19]. In contrast to the situation with MMS, we found that the loss of *RAD27* had no impact on YCS induced by Zeocin and that Pol32 loss reduces YCS, at least in cycling cells (Fig. 3). Whereas G1-arrested *pol32Δ* cells are not resistant to YCS, G1 cells lacking Trf4 and AP endonucleases are (Figs. 4, 5), suggesting that the redundancy among polymerases and lesion processing enzymes varies through the cell cycle. An alternative explanation might be that the nuclear actin levels and, therefore, actin-dependent remodeler activity show cell cycle variation. This remains to be explored.

A recent study in cultured mammalian cells has implicated nuclear actin polymerization in the downregulation of PrimPOL, an enzyme that combines primase and DNA polymerase activity and can load onto ssDNA to initiate DNA synthesis[32]. These authors argued that conditions that depolymerize actin (such a Latrunculin B) enable PrimPOL loading at stalled replication forks, which in turn prevents fork reversal and Rad51-mediated fork restart[32]. The authors proposed a mechanism of steric inhibition by nuclear F-actin filaments, yet they did not explore the possibility that nuclear actin may promote polymerase access through chromatin remodeling. Conclusions on how nuclear actin might control mammalian PrimPOL thus await further examination. Another mammalian study proposed that WASP, an actin polymerization factor, promotes RPA:ssDNA complex formation at stalled replication forks[34]. We do find Rfa1 and Rfa3 in our bicyclic peptide-actin pulldown and do not exclude the possibility that nuclear actin might influence RPA binding. However, we could not detect the yeast WASP homolog Las17 in the nucleus[46], making it unlikely that Las17 is a direct regulator of repair. We note that RPA-ssDNA stability might also be influenced by actin-containing nucleosome remodelers such as BAF or TRAPP.

The relevance of misprocessed BER intermediates and DSB induction is likely to have physiological relevance in mammals. Non-canonical BER processing occurs naturally in B lymphocytes during antibody class switch recombination (reviewed in ref. 2). In this case, cytosine deamination by the Activation Induced Deaminase (AID) enzyme, generates uracil lesions within the switch regions of the immunoglobulin heavy chain locus, which are rich in CpGs. These uracils are in close proximity and on opposite strands. They are subsequently processed by UNG, the uracil DNA glycosylase, and APE1, which generates ssDNA nicks that become DSBs[87]. The DSB triggers class switch recombination through an NHEJ-mediated translocation that alters the constant region associated with the expressed antibody. Intriguingly, a hyperactive and mistargeted form of AIDS is highly toxic as it leads to excessive DSBs, much like YCS[88]. Thus, what we present

here as a highly toxic misregulation of LP-BER upon depolymerization of cytoplasmic actin may, in fact, play a physiological role in rare but important contexts in humans.

## Methods

### Cell culture, strains, plasmids, and chemicals
Yeast strains and plasmids used in this study are listed in Supplementary Tables 1 and 2 and were derived from existing yeast plasmids[89,90]. Yeast cells were cultured in SC (synthetic complete-2% glucose) medium at 30 °C unless otherwise indicated. To deplete the auxin-dependent degron target, 0.5 mM indoleacetic acid (IAA, Sigma Aldrich) was added to the culture. The TORC1 and TORC2 inhibitors CMB4563 and NHP-BHS345 are closely related to imidazoquinolines, with CMB being roughly 10-fold more effective in the YCS assay. The compounds are dissolved in DMSO and were obtained from Stephen B. Helliwell (Novartis Institutes of Biomedical Research, Basel, Switzerland). In mammalian cells, they indiscriminately repress a broad range of PI3K-like kinases, while in yeast, they preferentially inhibit TORC2. All yeast strains, bicyclic peptides, and plasmids are available upon request to S.M.G., while requests for the imidazoquinolines should be addressed to S.B.H.

### CHEF gel analysis
Yeast genomic DNA was prepared in an agarose plug as described in the Instruction Manual (Bio-Rad, CHEF-DR II) with slight modifications. Yeast cells were spun and washed with ice-cold 50 mM EDTA-NaOH (pH 8.0), then the cell pellet was suspended in Zymolyase buffer (50 mM Na-phosphate pH7.0, 50 mM EDTA, 1 mM DTT) and embedded in a 1% agarose plug. Genomic DNA was prepared in the agarose plug by 0.4 mg/ml Zymolyase (20 T, Seikagaku) treatment in Zymolyase buffer at 37 °C for 1 h, followed by 1 mg/ml Proteinase K digestion in 10 mM Tris-HCl pH 7.5, 50 mM EDTA, 1% sodium N-lauroylsarcosinate at 50 °C more than 12 h, except genomic DNA preparation in Fig. 1 was performed at 30 °C. Chromosomal DNA was separated in non-denaturing 1% agarose gels in 0.5 x TBE on the CHEF-DR II Pulsed Field Electrophoresis Systems (Bio-Rad) as follows: 14 °C, 6 V/cm, 60 s switch time for 15 h, then 90 s switch time for 7 h. Chromosomal DNA was stained with SYBR-safe dye (Invitrogen) or occasionally where indicated, ethidium bromide or HDGreen Plus (INTAS). Imaging was generally on the Typhoon FLA 9500 scanner with LPB (510LP) filter (GE Healthcare) or, in early stages, the Chem Doc XRS system (Bio-Rad). Both show a linearity of dsDNA detection over 10'000 fold dilution. The gel data was then directly transferred into Image J software.

For CHEF gel quantitation, a rectangle covering an entire lane of a gel was set as ROI, and DNA intensity was plotted. Background from a flanking region of the gel without a sample was subtracted. We found no difference in monitoring intensity in Image J and the Typhoon scanner program, ImageQuant. In the Image J program, a rectangular region of interest (ROI) spanning from the largest chromosome band

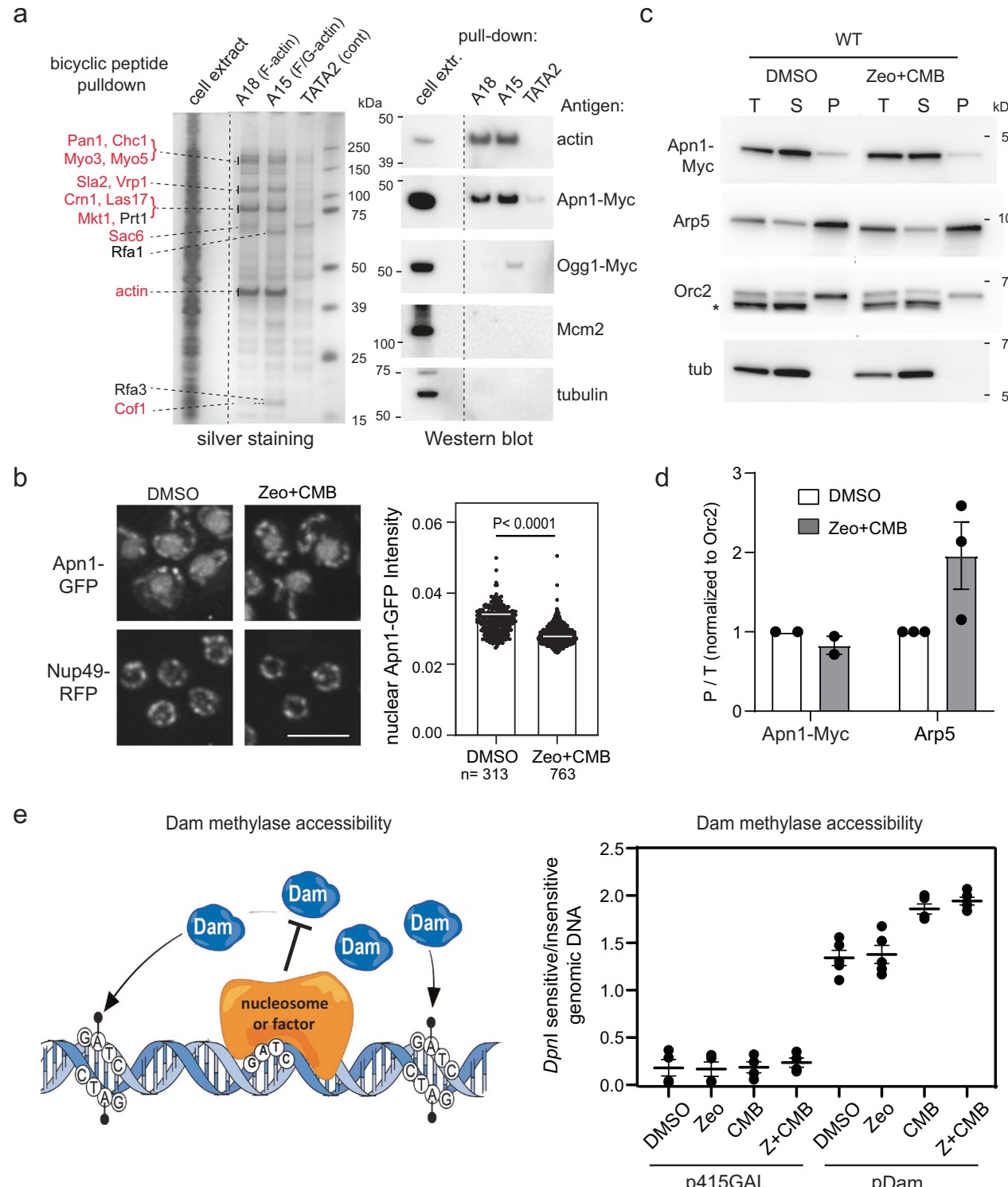

to ~20 kb in size was created, and the signal intensity was measured using line plot profiling. The same ROI was applied for each lane in a CHEF gel image. To measure B/A value, an ROI in each lane above 0.57 Mbp was created (above 0.57 Mbp includes Chr VIII/V bands through the largest Chr IV/XII) and one below 0.56 Mbp (extends from below Chr VIII/V to about 20 kbp). The latter includes chromosome fragments and the small Chr I, III, VI, and IX, even if intact. The integrated intensity above (A) and below (B) of the bar were measured. The same ROI was shifted in parallel to measure each lane in the gel. B over A was calculated and is indicated under each lane of each CHEF gel,

representing the degree of chromosome fragmentation. While absolute values are not directly comparable from gel to gel, within one gel, B/A value comparison is robust. All CHEF gels shown were quantified in a blinded manner (along with a limited number of repeats) and are presented in Supplementary Data 1. The quantitation values were not operator-specific.

Quantification of the mean number of breaks per chromosome was determined based on the assumption that DSBs are independently and uniformly distributed, i.e., they occur at the same frequency ($\lambda$) per unit length of DNA in all the chromosomes. Thus, the

**Fig. 7 | Enhanced Arp5 chromatin binding and DNA accessibility during YCS.**
**a** Apn1 and Ogg1 co-precipitate with actin. Bead-bound bicyclic peptides specific for F-actin (A18), F/G actin (A15)[75], or the control TATA2[75], were used to recover proteins from total yeast extracts. Proteins were visualized by silver staining, and prominent bands were identified by mass spectrometry from gel slices (see Methods, red = A15 and A18-pulldown, black = A15 only). The same fractions were analyzed by western blots for Apn1, Ogg1, Mcm2, and tubulin. Uncropped blots and mass spectroscopy results are in the Source data files. **b** Nuclear Apn1 levels drop slightly during YCS. *APN1-GFP NUP49-RFP* cells (GA-10504) cultured in SC were treated with DMSO ± 50 µg/ml Zeocin and 0.5 µM CMB4563 for 60 min. Spinning disc confocal images were captured of living cells in agarose plugs, and ImageJ quantified Apn1-GFP and Nup49-RFP intensities were plotted. Apn1-GFP was normalized to Nup49-RFP. Bar = 5 µm; n = cells imaged (n = 313, DMSO, and n = 763, Zeo + CMB). White bar = median; significance determined by Unpaired *t* test with Welch's correction, two-tailed; *p* < 0.0001. **c** INO80 subunit Arp5, but not Apn1, is slightly enriched on chromatin during YCS. APN1-9Myc tagged cells exponentially cultured in SC were treated with DMSO ± 50 µg/ml Zeocin and 0.5 µM CMB4563 for 80 min, and subjected to chromatin fractionation[72,75]. Total (T), soluble (S), and chromatin pellet (P) fractions were probed on western blots for Myc (9E10), Arp5, Orc2, and tubulin (see "Methods"). Full blots and quantitation in Source Data files. **d** P/T values for Apn1-Myc and Arp5 normalized to Orc2 are plotted, 1 = DMSO control. Anti-Orc2 cross-reacts with a 5kDa-smaller cytosolic protein (*). Uncropped blots are in Source data files. **e** Dam accessibility assay[78] monitors the relative accessibility of GATC motifs to ectopically expressed methylase (sketch modified from[99]). WT cells (GA-1981) carrying p415GAL (control) or p415GAL-Dam were treated with 50 µg/ml Zeocin, 0.5 µM CMB4563, or both for 80 min. Total genomic DNA was isolated, digested with *DpnI*, analyzed on a 1% agarose gel and stained by SYBR safe. *DpnI*-insensitive (intact genomic band) and *DpnI*-sensitive (below intact chromosomes) were quantified by ImageJ. Ratios from 5 biological replicas are plotted.

number of breaks in chromosome *i* follows a Poisson distribution of parameter $\lambda S_i$, where $S_i$ is the size of chromosome *i*. In this model, the mean fragment size over all the chromosomes is $\frac{S_{tot}}{\lambda S_{tot} + 16}$, where $S_{tot}$ is the length of the genome. Hence the mean number of breaks per chromosome can be calculated as a function of the mean fragment size (see Supplementary Fig. 2c legend). Given an estimated mean fragment size in the YCS experiments (i.e., maximal intensity distribution curve) of 100 ± 20 kb, the corresponding mean number of breaks per chromosome from Chr I to XVI are: 1.96; 6.92; 2.69; 13.04; 4.91; 2.30; 9.29; 4.79; 3.74; 6.35; 5.68; 9.18; 7.87; 6.68; 9.29; 8.07 (or 112 ± 22 total breaks; Fig. 1c, d). For an extended graph to 500 kb, see Supplementary Fig. 2c. A mean fragment size of 200–300 kb corresponds to a mean number of breaks genome-wide of 25–50, and a curve weighted for fragment size shows an approximate correlation of mean fragment size with predicted B/A values (Supplementary Fig. 2d). The theoretical relative intensity distribution of DNA fragments from randomly distributed DSBs is calculated as in reference [91].

## In vitro plasmid nicking assay
A pGEM13Zf(+) derived supercoiled plasmid[92] was incubated with Zeocin (Life Technologies) in MMR buffer (20 mM Tris-HCl pH = 7.6, 40 mM KCl, 5 mM MgCl₂, 50 ng/µl BSA, 1 mM glutathione) for 60 min at 37 °C. The sample was divided in two, then 0.002 U of recombinant hOGG1 (Trevigen, 4130-100-E) was added to one sample and further incubated for 30 min at 37 °C. Plasmid DNA was purified by NucleoSpin column (Macherey-Nagel), then analyzed on a 1% agarose gel containing RedSafe™ (Sigma-Aldrich) in 1 x TAE.

## Western blot and antibodies
Protein extracts were separated by SDS-PAGE or NuPAGE (Invitrogen) and transferred to nitrocellulose membranes BA-85 (Whatman) for probing. Antibodies used were: anti-actin (Millipore, Mab1501 clone C4) used 1:4000, anti-MCM2 (Santa Cruz, SC-6680) used 1:2000 dilution, rat anti-tubulin (YOL 1/34, ab6161, Abcam) used 1:10000 dilution, anti-tubulin (Thermo Fisher Scientific: MA1-80017), and anti-MYC (9E10) prepared in house, used 1:100 dilution, rabbit anti-histone H3 (polyclonal, ab1791, Abcam) used 1:1000 dilution, rabbit anti-histone H4 (polyclonal, ab10158, Abcam) used 1:5000 dilution, rabbit anti-Orc2 (Gasser laboratory, validated in ref. [93] used 1:1000 dilution, and anti-Rad53 (mouse monoclonal; 11G3G6 hybridoma clone, Gasser laboratory, validated in ref. [94] used 1:200. Rabbit anti-Arp5 polyclonal was a kind gift of Dr. M. Harata (Tohoku University, Japan), validated use at 1:10000 dilution[39]; rabbit anti-IAA17 was a kind gift of Dr. M. Kanemaki (NIG, Japan) used 1:1000 dilution. For bicyclic peptide pulldown, 10 µg of mouse anti-fluorescein (FITC) antibody (Jackson Immuno Res., clone IF8-IE4) was used for 50 µl of Protein G-Dynal beads (Thermo Fischer Scientific).

## Actin-bicyclic peptide and pull-down
Detailed information on the actin bicyclic peptides, including identification and characterization, are described in Gübeli et al.[75]. 10 µg of mouse anti-fluorescein (FITC) (Jackson Immuno Res., clone IF8-IE4) was pre-coupled with 50 µl of Protein G-Dynal beads (Thermo Fischer Scientific), to which 2 nmol of FITC labeled bicyclic peptides[75] A18 (ACREGQVACMVRKFECG), A15 (ACYRQWNKCENGWVRCG), and TATA2 (ANCPLVCAPRCR) were bound in yeast lysis buffer (50 mM HEPES pH7.5, 20 mM NaCl, 1 mM EDTA, 0.1% Triton X-100, protease inhibitor cocktail (Roche) for 1.5 h at room temperature, and washed three times before use. Yeast extracts were from exponentially growing yeast (~1 × 10⁷ cells/ml, 200 ml) washed once with ice-cold PBS, and then resuspended in 0.8 ml lysis buffer, and submitted to bead-beating with Zirconia beads, 6.5 Hz, 60 sec, 4 times at 4 °C. The cell lysate was clarified by centrifugation 12000 × g, 5 min at 4 °C, and 150 µl of total cell lysate was incubated with bicyclic peptide coupled Dynal beads (50 µl) for 1.5 h at 4 °C with constant rotation. After three rounds of washing in lysis buffer with 0.1% TritonX-100 and 20 mM NaCl at 4 °C, the proteins were eluted with 60 µl of 100 mM Glycine pH 2.0, 0.1% Triton X100. pH was quickly neutralized by 3.5 µl of 1 M Tris-HCl pH 9.5, and the protein sample was boiled in 1x NuPAGE sample buffer, and analyzed by NuPAGE followed by staining and Western blot. Bands were excised from the stained gel, and digested in situ for analysis by mass spectroscopy as follows. The gel slice is reduced with 10 mM TCEP, alkylated with 20 mM iodoacetamide, and cleaved with 0.1 µg porcine sequencing grade trypsin (Promega) in 25 mM ammonium bicarbonate (pH 8.0) at 37 °C for 16 h. The extracted peptides were analyzed by capillary liquid chromatography tandem mass spectrometry[95].

## Detection of chromatin accessibility by Dam
A yeast codon-optimized bacterial *Dam* was expressed from a single copy plasmid from a GAL1 promoter (p415-GAL1 backbone[89], see Supplementary Table 2). The transformants in a wild-type strain (GA-1981) were exponentially grown SC-LGG-leucine (synthetic medium minus leucine with 3% glycerol, 2% lactic acid, 0.05% glucose) to which 2% galactose was added for 30 min, and were then incubated with or without Zeocin and CMB as indicated at 30 °C. DNA was isolated with the MasterPure™ Yeast DNA Purification Kit (Lucigen). Isolated DNA was treated with *DpnI* (New England Biolabs) overnight at 37 °C, run on a 1% agarose gel stained by SYBR safe (Thermo Fisher Scientific), and visualized by Typhoon FLA 9500 scanner (GE Healthcare LifeSciences). This is a modified procedure based on ref. [78].

## Data reproducibility
All YCS experiments were repeated multiple times with similar results. The Typhoon scanned CHEF gels were quantified for DNA intensity distribution, and B/A values are in Supplementary Data 1. All gels were

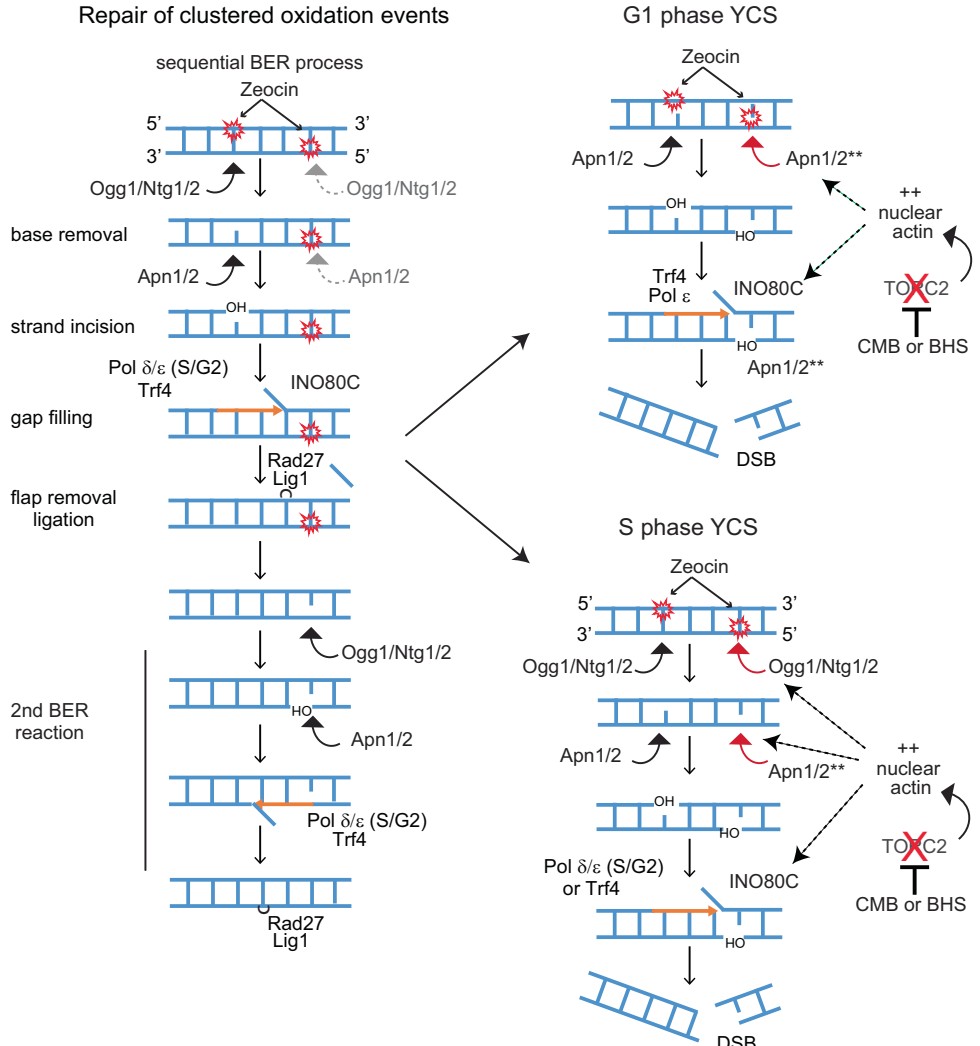

**Fig. 8 | Model of sequential BER of clustered damage and misregulation by nuclear actin.** We propose that the repair of clustered oxidization events on DNA becomes toxic in the presence of reagents that depolymerize actin in the cytoplasm, due to a miscoordination of the two-step BER process (see Discussion). The undisturbed step-wise repair of two closely positioned oxidation bases is shown on the left, with events that normally occur after the first lesion is repaired indicated in gray. Upon inhibition of TORC2, or disruption of cytoplasmic actin polymerization by other means, nuclear actin levels increase and provoke chromosome shattering in a manner dependent on Zeocin-induced damage and BER enzyme activity. In G1 phase cells, the increase in nuclear actin may activate Apn1/Apn2 directly or increase accessibility (see **) to drive premature processing of both lesions. In S-phase cells, nuclear actin may allow both glycosylases and AP endonucleases to prematurely process paired lesions. Unconstrained or enhanced processivity of DNA Pol δ in S phase, and Trf4 and DNA pol ε in both S and G1, contribute to DSB formation. The basic BER steps are original sketches based on recent reviews of BER pathways[3–5].

quantified in a blinded manner. Experiments that resulted in gels and Western blots (pull-downs, fractionation) were repeated at least twice and usually three times.

### Reporting summary
Further information on research design is available in the Nature Portfolio Reporting Summary linked to this article.

## Data availability
All original image captures of CHEF gels are available upon request to the corresponding author. Excel of CHEF gel quantitation for all gels shown and results discussed is provided as Supplementary Data 1. Source Data file containing uncropped Western blot images, relevant quantitation, and mass spectroscopy readouts is available. Reagents are available as described in Methods. Source data are provided in this paper.

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

## Acknowledgements

We dedicate this study posthumously to Sam Wilson (NIEHS), whose contributions to understanding base excision repair were enormous. We thank H. Araki, K. Shirahige, L. Maloisel, and S. Gangloff for yeast strains, M. Kanemaki for the degron-strain and IAA17 antibody, M. Harata for anti-Arp5, and C. Heinis for the cyclic peptides and advice. We thank Ragna Sack (FMI) for mass spec analyses of bicyclic peptide pull-down fractions, and Ireos Fillipuzzi (Novartis International, AG) for characterization of the TOR inhibitors. This work was supported in part by a Human Frontiers Science Program Grant RGP0017/2013, and the Swiss National Science Foundation grant 31003A-176286 (to S.M.G.) and 31003A-149989 (to J.J.). C.B.G. was supported by an FP7 Marie-Curie Intra-European Fellowship. We thank Gasser laboratory members for discussions, the Novartis Research Foundation for support, as well as Sam Wilson, Primo Schaer and Ulrich Hübscher for valuable insights.

## Author contributions

K.S. planned and carried out most of the experiments, helped write the paper, and led the project development, C.V.D.T. performed gel quantitation, S.B., C.B.G., B.V.L., and V.H. performed experiments, G.R. provided all the mathematical modeling of ss and ds break frequency, S.B.H. oversaw the collaboration on TOR inhibitors at Novartis Intl. AG and J.J. provided advice and supervised S.B. All authors read and commented on the manuscript. S.M.G. supervised the project, provided funding, helped plan experiments, and wrote and revised the manuscript.

## Competing interests

The authors declare that no competing interests exist.
