## [Transparent Peer Review file · Nature Communications]

TORC2 inhibition triggers yeast chromosome fragmentation through misregulated Base Excision Repair of clustered oxidation events

Corresponding Author: Professor Susan Gasser

Version 0:

Reviewer comments:

Reviewer #1

(Remarks to the Author)

This paper by Shimada et al. reports that low doses of the radiomimetic drug Zeocin, which directly causes low levels of DSBs along with closely spaced oxidized bases and AP sites on opposite DNA strands, generate an extreme accumulation of DSBs (yielding YCS) due to an uncoordinated activity of early onset BER enzymes in *S. cerevisiae* (with no link between actin depolymerization and DSB repair). This paper builds on past work by the Gasser lab showing that TORC2-mediated actin filament regulation is required for genomic stability following Zeocin or IR-induced DSBs (Mol. Cell, 2013). In that study, they showed that reduction (or deletion) of the canonical DSB repair pathways HR and NHEJ did not yield the YCS phenotype. In the current report, the authors show that eliminating BER glycosylases and/or endonucleases Atp1/2 and Rad1, inhibit DSB formation and YCS, indicating these BER enzymes play a direct role in generating the excess DSBs. It was also shown that Ogg1 and AP endonuclease co-precipitate with actin, possibly indicating that actin directly interferes with repair of Zeocin-induced damage.

Overall, this is a nice study and may merit publication in Nature Communications, after the concerns/comments given below are adequately addressed. A major concern with this work is the lack of quantitation of SBs from the gels for different cell strains (e.g., wt vs. mutant) or under different conditions. In addition, the last section on the interactions of BER factors with F-actin is less convincing since actin is known to bind a plethora of proteins, as stated by the authors. How specific are these interactions? Features such as binding strength and specificity are missing.

Specific concerns and comments are outlined below:

Specific Comments:

P. 5 (line 109) and P. 6 (lines 127-130): The ChEF gels are beautiful; however, they don't 'stand alone'. The authors should consider determining the ave. number of DSBs from these gels, as has been done in the past with other DNA break assays (e.g., see Bohr et al. Cell, 1985). Assuming a Poisson distribution for each chromosome number across the exposed cell population, one can determine the average number of DSBs/chromosome from the fraction free of DSB (P_0) using the Poisson equation $[-\ln(P_0)]$. This has worked well for numerous studies on NER in yeast minichromosomes or specific genes (e.g., see Hanawalt and Spivak. Nat Rev Mol Cell Biol. 2008 and refs. therein). Additionally, the authors should consider calculating the 'ensemble average' of breaks from the fragmented DNA profile (or 'DNA smear') to give the ave. overall break frequency/cell with each of the different DNA damaging agents (e.g., see Bessalov et al. Environ Mol Mutagen., 2001). Indeed, this might reveal a critical 'break frequency' that is needed to generate YCS.

P. 7 (line 147): This is a good assay for detecting low levels of SBs. Again, the authors should use it to quantify the number of SSB vs. DSB from these gels for accurate comparisons of the different agents (e.g., see Smerdon et al. Nucleic Acids Res., 1990).

P. 8 (line 162 and beyond): For more direct comparisons between wt and mutants (and between mutants), the authors should quantify the differences in overall strand breaks for different conditions; at least the percentage of breaks should be computed (e.g., see Czaja et al., DNA Repair, 2010).

P. 10 (line 228): Again, requires quantitation to make this statement

P. 12 (lines 260-261): Again, quantify

P. 13 (lines 281-283): Could it be that DNA pol and/or RNA pol elongation blockage induce enhanced AP endo activity (i.e. G1 vs log-phase)? Also, could the chromatin state could play a role? It's well known that chromatin structure regulates the activity of BER enzymes in vitro by preventing accessibility and processing of DNA damage (e.g., Prasad et al., MCB, 2007; Hinz et al., JBC, 2015; Maher et al. Nucleic Acids Res. 2019) and chromatin remodeling is required for sterically occluded DNA base damage (e.g., Bennett et al., Nucleic Acids Res. 2020). The more compact chromatin (on average) in G1-phase cells may partially 'release' BER enzymes from the 'hand-off' mechanism (e.g., Prasad et al., JBC, 2010) and APE activity may become more active.

P. 20 (lines 450-451): Could this be the case here? (The Mw of Bleomycin is almost 3X that of doxorubicin.) Quantitation of the number of SB induced by Zeocin, \pm CMB, would address this possibility.

Minor Comments:

P. 6 (line 118): Define MMS

P. 7 (line 138): "often" is a little strong-only about 10% of bleomycin-induced DNA lesions are bi-stranded, consisting of either two chemically identical breaks in opposite strands, or an abasic site with a closely opposed strand break (see ref. #19).

P. 7 (line 144): To clarify, insert "(Figure 1C, left panel)" after ... DNA backbone

P. 7 (line 145): Insert ", right panel" after (Figure 1C

P. 7 (line 147): Again, insert ", right panel" after (Figure 1C

P. 7 (line 157): Cite ref. for IAA inhibition of Ogg1

P. 9 (line 194-196): It appears that viability was determined from colony counting. Cell counting would give more accurate values and may be important for comparisons, as the differences are small. Also, the lack of correlation is another reason why break frequency/cell may be an important number.

P. 12 (line 273): Could be more if DSBs are clustered.

P. 15 (lines 336-337): Many proteins bind actin, so is this specific binding?. Other techniques may provide info. on the specificity of binding and mole ratio with these complexes (e.g., mass spec., size exclusion chromatography, sed. velocity, etc.).

P. 18 (line 399): reverse "the in"

P. 15 (lines 401-403): Clarify: As nicks would be processed 3'→5', the exo would not encounter the adjacent nick on the opposite strand.

P. 15-20: The Discussion section contains several overly long sentences (e.g., lines 360-363 and 375-378) and should be carefully gone over to improve grammar and clarity.

P. 19 (lines 430-431): This is indeed an interesting parallel.

P. 19 (lines 442-445): Interesting. What was the laser wavelength? Could CPDs have been induced at that λ ?

Reviewer #2

(Remarks to the Author)

In this manuscript, Shimada et al. report on a functional connection between the base excision repair (BER) pathway and actin dynamics. They follow up from their earlier observation (Shimada et al., Mol. Cell 2013) that inhibition of TOR2 signaling in budding yeast causes a hypersensitivity to the DNA-damaging drug Zeocin, manifested by chromosome shattering (YCS), and that this phenomenon is mediated by actin dynamics. Here they show convincing data that attribute the hypersensitivity to an inappropriate action of BER. Based on the notion that Zeocin tends to cause juxtaposed oxidative base lesions on opposite strands of the DNA, they argue that BER-mediated incisions by DNA glycosylases and/or AP endonucleases are responsible for introducing the double-strand breaks (DSBs). In line with the model that actin dynamics regulates BER activity, they observe direct interactions of human BER factors with actin.

This is an interesting manuscript that provides a plausible explanation for the phenomenon of chromosome shattering upon Zeocin treatment in combination with inhibition of actin polymerization. Although the conditions under which the phenotype is observable appear rather specialized, the study addresses a fundamental question: how BER events are coordinated to avoid interference with each other if they occur in close proximity. Use of Zeocin as an agent that exaggerates this general problem is therefore justified. The authors convincingly show that actin dynamics is important to prevent the harmful effects of BER under these conditions. That said, many questions remain unanswered by this study, and in light of these many loose ends, the model presented by the authors appears speculative at most. Additional experimental work therefore seems necessary to provide adequate support for the authors' conclusions and avoid over-interpretation of the data.

Major issues:

1. Neither the experiments nor the model provide any mechanism by which actin could regulate the activity of BER enzymes to make them differentiate between "single" and "paired" events. Do the authors wish to argue that a simple up- or down-regulation of BER activity would be sufficient to achieve this differentiation, or would they rather invoke a true interference mechanism that leads to an inhibition of new BER events in the vicinity of ongoing BER activity?

2. Related the first point: The interaction between BER factors and actin is poorly characterized, although it seems to be at the center of the authors' findings. Do they imply that the BER factors bind F-actin preferentially over G-actin? If so, this would need to be shown, but if not, I do not understand how a change in actin dynamics would regulate their activities. Use of the peptides is interesting, but doesn't exclude binding to either form of actin, and co-precipitation with F-actin doesn't disprove binding to G-actin or prove any differential binding. In addition, the peptides have apparently not yet been validated in a peer-reviewed study. The authors do not exclude other hypothesis, e.g. the sequestration of BER factors by actin outside of the nucleus. There are no experiments shown to address the localization or even the total amount of BER factors under YCS conditions.

3. The issue of cell cycle-specific requirements of AP versus NG is puzzling and unresolved. The authors show that G1 cells are much more sensitive than cycling cells to YCS conditions, and that deletion of the glycosylases (NG-delta) does not prevent YCS, whereas in cycling cells it does. This is puzzling because G1 cells make up a significant portion of a cycling cell population, so I would expect that some of the sensitivity of a cycling culture would be due to that of its G1 population. Are cells outside of G1 sensitive to YCS conditions at all? In order to control for this, cultures arrested outside of G1 should be analyzed. Moreover, the difference between G1 and non-G1 cells (labeled S phase without apparent reason) does not become clear in the model in Figure 6. Why should Ogg1/Ntg1/2 be insensitive to actin outside of G1?

4. Given that the Ogg1/Ntg1/2 are bifunctional enzymes and Ogg1 can produce breaks in the absence of any AP endonucleases in vitro, the requirement for AP endonucleases in vivo seems puzzling. This is not addressed here at all.

5. The use of CHEF gels for epistasis analysis appears questionable at best. There are no quantifications, which leaves less than black-and-white results open to interpretation as to whether a partial phenotype is closer to "epistatic" or to "additive". Moreover, the exposure time and Zeocin concentrations vary from experiment to experiment, which makes it very difficult to compare results from different gels/strains. And the concentration used in vitro was order(s) of magnitude lower than in vivo. Is this due to a lack of uptake by the cells?

6. Cell cycle experiments need associated FACS profiles.

7. The auxin experiments need western blots to control for protein degradation.

8. The effect of APN1/2 overexpression should also be verified by CHEF gels (Fig. S4C).

Minor issues:

9. How was viability measured (Fig. 2D)? The plate images of AP-delta do not reflect the numbers given below the plates, and they appear inconsistent with Fig. 4C.

10. In vitro interaction assays were done with human proteins – is there a reason why yeast proteins were not tested? In the accompanying manuscript, they show some evidence that the phenomenon may be conserved in human cells, but it would be helpful to generate data for yeast proteins as well.

11. Fig. 3C: please show both data points, not just the average.

Reviewer #3

(Remarks to the Author)

This submission by Shimada et al. follows up on published work by this group showing that combination of loss of TOR signaling and zeocin treatment in budding yeast results in chromosome shattering (YCS). The authors now investigate the role of BER in the process of generating the DSBs that yield YCS. They perform an extensive survey of BER genes implicated at distinct steps of this repair pathway. Overall this is a provocative observation. However, it is not entirely clear to what physiological process YCS might be relevant to. YCS is triggered in highly stressed cells lacking TOR signaling and therefore impaired in multiple physiological pathways. These cells have also limited viability. The conclusions rely heavily on CHEF gels without quantitative analysis of the signals and without analysis of multiple biological replicates (see below for details).

YCS occurs in cells treated with Zeocin but not with other compounds generating oxidative damage, suggesting that a large spectrum of lesions is required to start this process. In addition, TORC2 has to be inhibited. Yeast TORC2 orchestrates a complex stress response that regulates membrane lipids and proteins. The ability to prevent/rescue YCS with a variety of mutations in the BER pathway is interesting and provocative. However, the physiological significance of phenomenon is unclear. Indeed, some mutants rescue fragmentation but not viability, suggesting that it is a dead-end pathway. Moreover, what is the connection with TOR signaling? Is this conserved in higher eukaryotes? Chromosome fragmentation occasionally occurs in human tumor cells followed by rebuilding a scrambled chromosome during chromothripsis; YCS seems to be unrelated to this process.

A major technical problem with the manuscript is that it relies mostly on one experimental approach (CHEF gels) and there is no quantitative assessment of the data. This is a significant issue since: 1) these gels are not immune to variability (see examples below); 2) it leads the authors to employ various descriptors, which provide little information about the actual impact of the tested mutation and; 3) it also leads to over-interpretation of some data (see examples below). CHEF gels can

be scanned, as done in the Hurst et al. paper, the signal quantitated and average changes calculated with associated statistical significance.

Specific points:

Figure 1C: It seems that OGG1 alone induces some nicking in the absence of Zeocin, which complicates the interpretation of the data.

There is some inherent variability between CHEF gels showing the same experiment, emphasizing the need for replicates and quantitation. Compare Figure 2B, lane 17 with Figure S1B, lane 10 (rescue by *ogg/ntg* mutations). Rescue in Supp Figure 1B is less convincing.

Lane 179 and Figure 2C: "almost completely", please quantitate.

More consistency with the experimental design would have made the experiments easier to compare. For example, it is difficult to compare the impact of the double vs. triple mutants *apn1*, *apn2*, *rad1* between Figure S1C (BHS, Zeocin 75uM, 45 and 90 minutes) with Figure S1D (CMB, Zeocin 50uM, 60 minutes). There are more instances in which the authors use CMB or BHS without explanation. Since the Hurst submission indicates that CMB is more efficient at triggering YCS and more specific towards TORC2, why not use CMB for all experiments?

Figure 3A and 3B. It looks as if the double *pol4/pol32* mutant rescues better than *pol32* alone: quantitation should tell. Lane 224-225: "very minor resistance": I cannot see a difference between lane 4 and lane 8. Also, in this experiment it seems that zeocin alone triggers some fragmentation: compare lanes 1 and 2 and lanes 5 and 6. Similarly, lane 228 figure 3C: "a minor resistance".

Lanes 240-241, Figure S2B: "almost identical". It seems that the *pcd1* mutant might have more low molecular weight DNA.

Lane 250, Figure 4A: "little or no resistance". Please quantitate.

Figure 2B shows a strong rescue by *ogg1/ntg1/ntg2* triple mutant in exponentially growing cells: compare lane 9 and 18. In contrast, Figure S3A shows poor rescue in exponentially growing cells, compare lane 4 to lane 8. This raises two important issues: 1) inconsistencies between experiments, which can only be addressed by performing multiple biological replicates and quantification and 2) it does not support the conclusion that there is a difference between G1 arrested and exponentially growing cells (S3A) as there is very little, if any, impact in exponentially growing cells between WT and triple mutant.

Compare second lane of Figure 4A and second lane of Figure 4B: this is another example showing that the impact of zeocin alone varies between experiments.

Lane 269-270 and Figure 4D: "*cdc9*... generated fragmentation on zeocin even in the absence of TORC2 inhibition". I assume that this statement is based on the comparison of lane 2 and lane 6 (Figure 4D). These samples look nearly identical, which does not support the conclusion.

To monitor actin-BER interactions, the author elected to use actin-binding peptides described in a preprint. If I understand properly, the rationale to use G- and F-actin specific ligands would be to identify actin-binding proteins that interact specifically with polymerized actin. However, the protein gel shows that all identified proteins bind to both peptides, with the notable exception of cofilin, which should bind to both F- and G-actin but does not interact with the F-actin specific peptide.

With regards to the direct interaction between actin and BER proteins in human cells, the author monitor association in the pellet following high-speed centrifugation. However, polymerized actin filaments could trap other proteins and drag them into the pellet in a non-specific manner, dependent on the concentration of proteins. These experiments do not seem to have been controlled for total protein concentration, a factor that strongly influence the ability of proteins to pellet. Actin-containing samples have a protein concentration several fold higher than the control. It might have made more sense to pull down actin in solution and monitor binding.

Typos..

Lane 57: one-ended parenthesis.

Lane 247: one-ended parenthesis.

Lane 251: remove "cf".

Lanes 258-259: please check lane 258. Also, this should be rewritten in a way that does not suggest *pol4* mutant had an impact on exponentially growing cells.

Version 1:

Reviewer comments:

Reviewer #1

(Remarks to the Author)

In the revised manuscript, Shimada et al. have successfully addressed many of my concerns/comments. Unfortunately, there are still issues with the new version, particularly with the quantitation methods they have used. In the following list, I have gone through the rebuttal comment by comment:

1) ...A major concern with this work is the lack of quantitation of SBs from the gels for different cell strains (e.g., wt vs. mutant) or under different conditions.

(p.24, 1st par.) As implied by the authors, the photo intensities are linear only over a small range and don't correlate with the actual pixel intensities of the scans. However, the Typhoon FLA 9500 response is linear over 5 orders of magnitude and Bio Rad chemiDoc XRS system has a dynamic range of 4 orders of magnitude. Therefore, one doesn't have to worry about linearity over the changing gel scan area due to changes in exposure.

Unfortunately, since the ImageJ program was used for quantitation of gel images, I assume screen shots of the data were analyzed. Because of the reason stated above, the analysis is more accurate when the file export information is directly analyzed. For the Typhoon data, this would be with the ImageQuant program. As stated below, the key is to obtain the number-average length (L_n).

2)Indeed, this might reveal a critical 'break frequency' that is needed to generate YCS.

(p.24, 2nd par.) The ratio chosen (B/A) is systematically flawed because the A region (>0.57 Mbp) contains long SSB fragments, and the amount changes with dose. Thus, this ratio would give a nonlinear dose-response (SSB/ μg vs [Zeo]).

There are several papers that address the quantitation of these types of gels [e.g., Czaja et al. DNA Repair 9(9):976, 2010; Li et al. Sci. Reports 11(1):18393, 2021]. The background of undamaged chromosomes on CHEF gels can be subtracted to give a 'smear' which allows for determination of the ensemble average, the median length, & the number average length (L_n). It follows that $\text{SSBs}/(\text{unit DNA}) = 1/L_n (+\text{Zeo}) - 1/L_n (-\text{Zeo})$.

3) ... the authors should use it to quantify the number of SSB vs. DSB from these gels for accurate comparisons of the different agents.

OK

4) authors should quantify the differences in overall strand breaks for different conditions; at least the percentage of breaks should be computed.

Insufficient response -See comments above (#1 & #2)

5) P. 10 (line 228): Again, requires quantitation to make this statement.

Insufficient response -See comments above (#1 & #2)

6) P. 12 (lines 260-261): Again, quantify.

Insufficient response -See comments above (#1 & #2)

7) P. 13 (lines 281-283): Could it be that DNA pol and/or RNA pol elongation blockage induce enhanced AP endo activity (i.e. G1 vs log-phase)?

OK

8) The more compact chromatin (on average) in G1-phase cells may partially 'release' BER enzymes from the 'hand-off' mechanism (e.g., Prasad et al., JBC, 2010) and APE activity may become more active.

It is confusing to tell if P/T panel goes with 7B or 7D. This should be clarified.

9) Quantitation of the number of SB induced by Zeocin, \pm CMB, would address this possibility.

OK

Minor Comments

All OK

Minor Comments on new version:

p. 6 (lines 153-153): The statement "...nor is there any sign of DSB formation..." doesn't align with the data. Clarify.

p. 17 (line 490): Delete "nuclei"

p. 17 (line 496): ...subunit of [the] INO80 complex, which [is] an actin nucleosome.....

p. 24 (lines 720-722): Sentence is not clear and should be rewritten.

Reviewer #2

(Remarks to the Author)

In their revised version, the authors have addressed many of the reviewers' original concerns and have strengthened their arguments substantially. This has resulted in a more coherent and convincing study.

The effect of Zeocin on DSB formation via the BER pathway is now very well documented.

The link to the actin system via the INO80 complex is again plausible, but still speculative because relevant functional data (e.g. effects of ino80 mutants on YCS) are not provided. Such data appear to be in revision in an additional manuscript (Hurst et al) that the authors mention but do not show; including them in this manuscript would have helped the authors to make their point. Nevertheless, I am mostly happy with how the authors have responded to the reviewers' comments.

Version 2:

Reviewer comments:

Reviewer #1

(Remarks to the Author)

The revised manuscript by Shimada et al. have adequately addressed all my previous concerns/comments. I have made some comments below (just for the authors' edification) to individual responses. All the other responses that aren't mentioned below were fine. Furthermore, the additional data (e.g., the demonstration of enhanced DNA accessibility with CMB or CMB plus Zeocin) have markedly improved the quality and completeness of this manuscript. Therefore, I think this manuscript is ready for publication in Nature Communications.

Responses to:

- 1) "We have now provided extensive Note that because B/A is an "internal lane" ratio, this is still valid."
 - Good job - Fig. 2 (for reviewers) is convincing
- 2) "...we are not scanning photographs"
 - Noted and clearly emphasized
- 3) "...measure B/A value, a division of the lane above and below 0.57 Mbp was created ss gaps or nicks in the chromosome fragments is not relevant, nor is it scorable with nondenaturing gels."
 - Although altered migration on native gels with clustered SSBs (and the differential dependence the frictional coefficient can have with different chromosome sizes) could make this number 'relevant'; however, I think the authors have now adequately considered this issue
- 4) "... intensity of ds DNA..... representing the degree of chromosome fragmentation. We are convinced that the B/A value comparison is robust and monitors irreparable DSBs which is what we aim to monitor."
 - Yes, and the ratio is now better explained
- 5) "We also modeled B/A ratios based on the mean fragment size detected by CHEF gels, and the values (see Supplemental Figure 2D) fit our measurements well."
 - Good addition (especially correlation curve)
- 6) Authors response to 4) authors should quantify the differences in overall strand breaks for different conditions; at least the percentage of breaks should be computed.
 - Meant % of total DNA (i.e., DSB/ μ g) which was adequately addressed earlier
- 7) "... we assume that DSBs are independently and uniformly distributed, i.e. they occur at the same frequency ($\lambda\lambda$) per unit length of DNA in all the chromosomes"
 - The Poisson distribution only assumes that the frequency is constant across each set of chromosomes and could vary between each set of chromosomes. (Indeed, this is quite possible since each set has different sequences.)
- 8) "The more compact chromatin (on average) in G1-phase cells may partially 'release' BER enzymes from the 'hand-off' mechanism (e.g., Prasad et al., JBC, 2010) and APE activity may become more active."
 - Yes, meant stationary phase

REVIEWER COMMENTS on Shimada et al (Replies in blue)

Reviewer #1 (Remarks to the Author):

This paper by Shimada et al. reports that low doses of the radiomimetic drug Zeocin, which directly causes low levels of DSBs along with closely spaced oxidized bases and AP sites on opposite DNA strands, generate an extreme accumulation of DSBs (yielding YCS) due to an uncoordinated activity of early onset BER enzymes in *S. cerevisiae* (with no link between actin depolymerization and DSB repair). This paper builds on past work by the Gasser lab showing that TORC2-mediated actin filament regulation is required for genomic stability following Zeocin or γ IR-induced DSBs (Mol. Cell, 2013). In that study, they showed that reduction (or deletion) of the canonical DSB repair pathways HR and NHEJ did not yield the YCS phenotype. In the current report, the authors show that eliminating BER glycosylases and/or endonucleases *Apn1/2* and *Rad1*, inhibit DSB formation and YCS, indicating these BER enzymes play a direct role in generating the excess DSBs. It was also shown that *Ogg1* and AP endonuclease co-precipitate with actin, possibly indicating that actin directly interferes with repair of Zeocin-induced damage.

Overall, this is a nice study and may merit publication in Nature Communications, after the concerns/comments given below are adequately addressed. **A major concern with this work is the lack of quantitation of SBs from the gels for different cell strains (e.g., wt vs. mutant) or under different conditions.**

We have addressed this by quantifying all gels and other quantitative assays. It is difficult to compare direct values among independent experiments, largely because Zeocin does not always have exactly the same specific activity per μ g. This is why we *always* do the WT or control strains in parallel, and always test Zeocin alone, CMB or BHS alone, as well as the mixture, and run all samples at the same time. When assays are performed in parallel the results are highly reproducible. Therefore we present a “typical” gel with quantitation of fragmented chromosomes over non-fragmented chromosomes for each experiment (splitting the intact vs fragmented (< 0.5 Mb) was more quantitative and reproducible than integrating absolute area beneath a curve of a gel line-scan, as the latter is highly dependent on the exposure). The method of quantitation is explained in the Methods section.

In addition, the last section on the interactions of BER factors with F-actin is less convincing since actin is known to bind a plethora of proteins, as stated by the authors. How specific are these interactions? Features such as binding strength and specificity are missing.

We agree that the *in vitro* binding studies were not highly quantitative and are difficult to control. Thus we omit the interaction assays using purified human proteins and only show the actin-bicyclic peptide pulldown out of whole yeast extracts, since it is clear by mass spec that we were pulling out the abundant bona fide actin binding proteins (as expected), and found only a subset of nuclear (repair) proteins binding actin. Binding strength is very difficult to measure. Actin is a difficult ligand to do biophysical analysis with (it forms filaments and precipitates *in vitro*). Extensive attempts to get quantitative interaction values with purified *Ogg1* and actin were inconclusive, while actin and *Arp4* interaction is well characterized and crystal structures are published (Cao T, et al. Crystal structure of a nuclear actin ternary complex. PNAS USA. 2016;113:8985–8990. doi: 10.1073/pnas.1602818113).

Specific concerns and comments are outlined below:

Specific Comments:

P. 5 (line 109) and P. 6 (lines 127-130): The CHEF gels are beautiful; however, they don't 'stand alone'. The authors should consider determining the ave. number of DSBs from these gels, as has been done in the past with other DNA break assays (e.g., see Bohr et al. Cell, 1985). Assuming a Poisson distribution for each chromosome number across the exposed cell population, one can determine the average number of DSBs/chromosome from the fraction free of DSB (P_0) using the Poisson equation $[-\ln(P_0)]$. This has worked well for numerous studies on NER in yeast minichromosomes or specific genes (e.g., see Hanawalt and Spivak. Nat Rev Mol Cell Biol. 2008 and refs. therein). Additionally, the authors should consider calculating the 'ensemble average' of breaks from the fragmented DNA profile (or 'DNA smear') to give the ave. overall break frequency/cell with each of the different DNA damaging agents (e.g., see Bernal et al. Environ Mol Mutagen., 2001). Indeed, this might reveal a critical 'break frequency' that is needed to generate YCS.

Thanks for these comments. We have carefully quantified all CHEF gels, which indeed strengthens our conclusions. We also calculated the number of DSBs detected in our assays using a similar random break distribution model as that mentioned. Taking into account the size of yeast chromosomes and the average size of the "smear" which indicates the distance between of DSBs, we find between 80 and 120 DSBs per cell are necessary (theoretically) to generate a Gaussian distribution of fragments centered around 100kb in length. Since Povirk earlier proposed that Zeocin and related molecules (Bleomycin) generate 10 x more ss lesions than DSBs (in wt conditions), we can expect that there are more like 800 – 1200 oxidation events that occur per cell, under YCS conditions (during the 60 min incubation). Thus 50-80 microgram per ml of Zeocin is sufficient to provoke this level of damage, and in wild-type cells in the absence of TORC2 inhibitor, these are efficiently repaired without cell cycle block. We find no YCS in the absence of TORC2 inhibition at the level of Zeocin used. Please note that H₂O₂ also generates 8-oxoG which is a substrate for BER, but it does not lead to YCS, even on CMB/BHS, which we believe argues that the clustering of Zeo-induced lesions is key to the observed fragmentation. Others have highlighted the crucial importance of having mechanisms to deal with clustered damage. We believe YCS occurs because of a failure to coordinate repair of clustered or adjacent oxidation events. Since we can trigger this by increasing G actin levels in the nucleus (new Figure 6) we argue that nuclear actin interferes with stepwise BER repair processes.

P. 7 (line 147): This is a good assay for detecting low levels of SBs. Again, the authors should use it to quantify the number of SSB vs. DSB from these gels for accurate comparisons of the different agents (e.g., see Smerdon et al. Nucleic Acids Res., 1990).

Done. The molar ratio of Zeo to plasmid in this assay is roughly 1:1 at the lowest concentration of Zeocin. We observe almost complete relaxation of the sc plasmid, but only 10% is linearized upon addition of Ogg1. This is in line with the Povirk data. With a 10-fold molar excess of Zeo we observe 35% conversion to linear DNA and at 100 fold excess only 20% remains nicked, the rest is either linearized or starts to show degradation. Given that Ogg1 activity is not 100% efficient (nor is Zeocin), our results are roughly the same as Povirk's (1996). H₂O₂ and MMS do not show the same pattern of damage as Zeocin in vivo.

P. 8 (line 162 and beyond): For more direct comparisons between wt and mutants (and between mutants), the authors should quantify the differences in overall strand breaks for different conditions; at least the percentage of breaks should be computed (e.g., see Czaja et al., DNA Repair, 2010).

See above – we have now quantified all CHEF gels for a more accurate comparison of conditions and mutants.

P. 10 (line 228): Again, requires quantitation to make this statement
We agree; quantitation supports our conclusion of a weak difference.

P. 12 (lines 260-261): Again, quantify Done and incorporated into text

P. 13 (lines 281-283): Could it be that DNA pol and/or RNA pol *elongation blockage* induce enhanced AP endo activity (i.e. G1 vs log-phase)? We find that loss of the 3 glycosylases (NG Δ) does not prevent YCS in G1, but does have an effect in log phase cells. So one interpretation is that AP endonuclease is processing without glycohydrolase activity, generating nicks that get converted to DSBs, despite NG ablation. The reviewer asks if either RNA or DNA polymerase blockage enhances AP activity in G1... we showed in Shimada et al., Mol Cell 2013 that the addition of alpha-amanitin (RNA PolII inhibitor) did not induce nor prevent YCS, nor does *RPO21* depletion (RNA-PolII largest subunit). Moreover, since slowing DNA polymerases (*pol32 Δ* , *pol3-ct*, *pol2-18*) reduces YCS, we do not think that DNA pol blockage actually induces AP activity.

Also, could the chromatin state could play a role? It's well know that chromatin structure regulates the activity of BER enzymes in vitro by preventing accessibility and processing of DNA damage (e.g., Prasad et al., MCB, 2007; Hinz et al., JBC, 2015; Maher et al. Nucleic Acids Res. 2019) and chromatin remodeling is required for sterically occluded DNA base damage (e.g., Bennett et al., Nucleic Acids Res. 2020). The more compact chromatin (on average) in G1-phase cells may partially 'release' BER enzymes from the 'hand-off' mechanism (e.g., Prasad et al., JBC, 2010) and APE activity may become more active.

We do very much agree that chromatin status plays a major role in regulating the repair of adjacent lesions, and we present evidence that YCS conditions (actually TORC2 inhibition itself) augments chromatin accessibility as measured by DAM modification (free DAM accessibility assay in Figure 7). We find (to be published in Hurst et al, in revision) that INO80 mutants are resistant to YCS and show here that Arp5, a core component of INO80C is slightly enriched on chromatin under YCS conditions. It is known that the Arp4-actin subcomplex of INO80C is rate limiting for nucleosome remodeling (e.g. Brahma, S. et al. The Arp8 and Arp4 module acts as a DNA sensor controlling INO80 chromatin remodeling. Nat Commun 9, 3309 (2018). <https://doi.org/10.1038/s41467-018-05710-7>, and others, cited in our ms), thus we propose that nuclear actin actually stimulates INO80, driving premature opening and processing of adjacent oxidative damage. This is the take home of the revised Figure 7, which includes evidence that chromatin is more accessible under YCS conditions. The enhanced processivity of DNA polymerases may be a direct outcome of chromatin alternations. The misregulation of Arp1 could also be chromatin linked or Arp1 could be a direct target of nuclear actin, but we do not see more Arp1 binding under YCS conditions. Interestingly, we do see more Ogg1. The concept that nuclear actin affects chromatin accessibility is now discussed at length in our paper and is incorporated into the model (Figure 8).

P. 20 (lines 450-451): Could this be the case here? (The Mw of Bleomycin is almost 3X that of doxorubicin.) Quantitation of the number of SB induced by Zeocin, \pm CMB, would address this possibility.

We determined 8-oxoG rates induced by Zeocin and show that these are nearly identical \pm CMB (now included as Suppl Figure 2b). They were also similar to that induced by the level of H₂O₂ used in Fig. 1. Thus we do not think that Zeocin uptake is a major rate limiting target for CMB inhibition, at least not in our assay. In addition, we could detect YCS with γ IR (Shimada et al., Mol Cell 2013), which has no “uptake” issues. The problem with using γ IR is that ionizing radiation oxidizes other molecules besides DNA and the irradiation level of a population of yeast cells is difficult to control (they do not grow in monolayers).

Minor Comments:

P. 6 (line 118): Define MMS Done

P. 7 (line 138): “often” is a little strong—only about 10% of bleomycin-induced DNA lesions are bi-stranded, consisting of either two chemically identical breaks in opposite strands, or an abasic site with a closely opposed strand break (see ref. #19). This has been modified to be more quantitative; we agree that 10% is not often

P. 7 (line 144): To clarify, insert “(Figure 1C, left panel)” after ... DNA backbone Done. The figure numbers have changed but we have clarified this by referring to the sketch.

P. 7 (line 145): Insert “, right panel” after (Figure 1C) Done. The figure numbers have changed but we have clarified this by referring to the sketch.

P. 7 (line 147): Again, insert “, right panel” after (Figure 1C) Done. The figure numbers have changed but we have clarified this by referring to the sketch.

P. 7 (line 157): Cite ref. for IAA inhibition of Ogg1

This is a misunderstanding: IAA does not inhibit Ogg1. We have tagged Ogg1 endogenously with a degron tag that is stimulated by IAA (a plant hormone) leading to its rapid degradation. We include controls for Ogg1 and Rad1 degradation as requested by reviewer 3.

P. 9 (line 194-196): It appears that viability was determined from colony counting. Cell counting would give more accurate values and may be important for comparisons, as the differences are small. Also, the lack of correlation is another reason why break frequency/cell may be an important number.

Viability is determined by serial drop assay in triplicate (minimally triplicates, sometimes many more replicates are done). Quantitation by rigorous colony counting after fixed periods of growth is a long-validated assay in the yeast community. The confusion perhaps arose because the culture plate shown was a wrong (figure assembly error), and we have corrected this and this data is now in Figure 4.

P. 12 (line 273): Could be more if DSBs are clustered.

It is unclear to us why multiple DSBs would be clustered, but it might occur. Importantly, our estimate is stated as a minimal estimate.

P. 15 (lines 336-337): Many proteins bind actin, so is this specific binding?. Other techniques may provide info. on the specificity of binding and mole ratio with these complexes (e.g., mass spec., size exclusion chromatography, sed. velocity, etc.).

We answered this above: specificity in the pulldown assay is only based on the facts that other proteins in the assay did not bind actin, and the proteins identified were not recovered with an unrelated bicyclic peptide. Note that we have removed the section on “F-actin precipitation of purified proteins” exactly because it is difficult to determine specificity when using actin precipitation as an assay. We leave in the yeast-extract actin-pulldown (which reflects binding to a bicyclic peptides rigorously selected (see ACS Chem Biol 2021 May 21;16(5):820-828. doi: 10.1021/acscchembio.0c00825). Moreover, we confirm by mass spec that the most abundant proteins recovered are the known actin regulators of the cytoplasm, yet we detect robust Apn1 binding and RFA binding (and weakly Ogg1), with many negative controls being other nuclear repair proteins. An accompanying paper addresses the relevance of various actin-binding and filament-forming mechanisms in YCS (Hurst et al., in revision), and again this study implicates actin-containing nucleosome remodelers in the misregulation of repair.

P. 18 (line 399): reverse “the in” Done

P. 15 (lines 401-403): Clarify: As nicks would be processed 3'->5', the exo would not encounter the adjacent nick on the opposite strand. We have clarified this in a sketch in the model

P. 15-20: The Discussion section contains several overly long sentences (e.g., lines 360-363 and 375-378) and should be carefully gone over to improve grammar and clarity. We have taken care to shorten and simplify our sentences

P. 19 (lines 430-431): This is indeed an interesting parallel. We agree and we use this to indicate the potential relevance of our findings for other systems

P. 19 (lines 442-445): Interesting. What was the laser wavelength? Could CPDs have been induced at that λ ? The fact that actin depolymerization affects XRCC1 recruitment to laser-induced damage (without actin itself showing up at the lesion) is now published (Hurst et al., MBC 2021). The laser used was 405 nm- wavelength, which could create CPD lesions. This is mentioned in the discussion.

Reviewer #2 (Remarks to the Author):

In this manuscript, Shimada et al. report on a functional connection between the base excision repair (BER) pathway and actin dynamics. They follow up from their earlier observation (Shimada et al., Mol. Cell 2013) that inhibition of TOR2 signaling in budding yeast causes a hypersensitivity to the DNA-damaging drug Zeocin, manifested by chromosome shattering (YCS), and that this phenomenon is mediated by actin dynamics. Here they show convincing data that attribute the hypersensitivity to an inappropriate action of BER. Based on the notion that Zeocin tends to cause juxtaposed oxidative base lesions on opposite strands of the DNA, they argue that BER-mediated incisions by DNA glycosylases and/or AP endonucleases are responsible for introducing the double-strand breaks (DSBs). In line with the model that actin dynamics regulates BER activity, they observe direct interactions of human BER factors with actin.

This is an interesting manuscript that provides a plausible explanation for the phenomenon of chromosome shattering upon Zeocin treatment in combination with inhibition of actin polymerization.

Although the conditions under which the phenotype is observable appear rather specialized, the study addresses a fundamental question: how BER events are coordinated to avoid interference with each other if they occur in close proximity. Use of Zeocin as an agent that exaggerates this general problem is therefore justified. The authors convincingly show that actin dynamics is important to prevent the harmful effects of BER under these conditions. That said, many questions remain unanswered by this study, and in light of these many loose ends, the model presented by the authors appears speculative at most. Additional experimental work therefore seems necessary to provide adequate support for the authors' conclusions and avoid over-interpretation of the data.

Major issues:

1. Neither the experiments nor the model provide any mechanism by which actin could regulate the activity of BER enzymes to make them differentiate between "single" and "paired" events. Do the authors wish to argue that a simple up- or down-regulation of BER activity would be sufficient to achieve this differentiation, or would they rather invoke a true interference mechanism that leads to an inhibition of new BER events in the vicinity of ongoing BER activity?

We include several new experiments to address this question: first we use a polymerase delta elongation mutant (*pol3-ct*, lacking just a few aa at the C terminal end of the polymerase), and find partial YCS resistance, arguing that polymerase elongation does drive at least a subset of the fragmentation. We also rule out that actin-depolymerization (TORC2 inhibition) leads to an increase of APN1 on chromatin. Finally, in an accompanying paper we show that INO80 activity is necessary for the YCS effect (Hurst et al., in revision). Here we show that Arp5, a unique INO80C subunit, shows increased chromatin binding under YCS conditions. We rule out a lot of other potential players as being directly affected by the TORC2 inhibition (and the ensuing increase in nuclear actin). Finally, knowing that INO80 triggers histone loss under DNA damaging conditions (Hauer et al., NSMB 2016; Cheblal et al., Mol Cell 2020) we propose that chromatin accessibility enhances DNA polymerase processivity. Here we confirm that YCS conditions increase chromatin accessibility genome-wide (new Figure 7E). We do not rule out that Apn1 activity might also be enhanced, but we rule out that its abundance in the nucleus is rate limiting through a number of overexpression assays (including the expression of a NLS-Apn1 fusion). Because TORC2 inhibitors do not increase 8-oxoG levels provoked by Zeocin (Suppl figure 2) nor affect Rad52 focus formation (Shimada et al., 2013), the best hypothesis is misregulation of BER by increased nuclear actin. That is our model and we believe our data argue effectively for this. Our quantitative analysis of the frequency of lesions helps support this interpretation.

2. Related the first point: The interaction between BER factors and actin is poorly characterized, although it seems to be at the center of the authors' findings. Do they imply that the BER factors bind F-actin preferentially over G-actin? If so, this would need to be shown, but if not, I do not understand how a change in actin dynamics would regulate their activities. Use of the peptides is interesting, but doesn't exclude binding to either form of actin, and co-precipitation with F-actin doesn't disprove binding to G-actin or prove any differential binding. In addition, the peptides have apparently not yet been validated in a peer-reviewed study. The authors do not exclude other hypothesis, e.g. the sequestration of BER factors by actin outside of the nucleus. There are no experiments shown to address the localization or even the total amount of BER factors under YCS conditions.

Actually we do not base our proposed mechanism on the direct action of actin on repair factors – although we cannot exclude that Apn1 might be a target of nuclear actin, and therefore it is included as an option in our model. In mammalian cells Ape1 actually accumulates in the cytosol under YCS conditions (Hurst et al., MBC 2021), and we find in yeast that increased G actin levels in the nucleus do not increase Apn1 levels on chromatin (Figure 7). Even expression of an NLS-Apn1 only slightly increases YCS rates, and definitely does not confer resistance under YCS conditions, thus despite the clear importance of AP endonucleases in chromosome shattering, their abundance does not seem to be the crucial factor that TORC2 inhibition controls. Moreover, despite many attempts, we could not detect any significant increase in Ogg1 or Apn1 activity *in vitro* upon the addition of actin (data not shown). *In vitro* assays suffer from the fact that actin polymerizes spontaneously in physiological (or even subphysiological) levels of salt, thus most repair assay conditions lead to actin polymerization and precipitation. We therefore focused on the DNA polymerases involved in LP-BER (and SP-BER in yeast), raising the question whether their processivity affects shattering (indeed it does; Figures 3 and 5).

A recent publication argues that nuclear actin reduces the activity of PRIMPOL in mammals (Palumbieri et al., Nuclear actin polymerization rapidly mediates replication fork remodeling upon stress by limiting PrimPol activity. Nat Commun 14, 7819, 2023), yet yeast does not have PRIMPOL, nor nuclear actin filaments (as far as we can tell). In the discussion we explain that we have quite the opposite effect: we find a detrimental effect of nuclear actin due to enhanced DNA polymerase processivity, they argue for a positive role of nuclear F-actin in *protecting* from this. While we cannot completely exclude direct effects of nuclear actin filaments at damage, we prefer the better substantiated model that polymerase processivity is influenced by nucleosome remodelers that require actin to bind nucleosomes (e.g. INO80 in yeast, BRG1 possibly in mammals; Cohen et al., 2020; <https://doi.org/10.1093/nar/gkq559>, Shimada et al., 2008; <https://doi.org/10.1016/j.cub.2008.03.049>). This has been amply documented by our laboratory and others.

3. The issue of cell cycle-specific requirements of AP versus NG is puzzling and unresolved. The authors show that G1 cells are much more sensitive than cycling cells to YCS conditions, and that deletion of the glycosylases (NG-delta) does not prevent YCS, whereas in cycling cells it does. This is puzzling because G1 cells make up a significant portion of a cycling cell population, so I would expect that some of the sensitivity of a cycling culture would be due to that of its G1 population. Are cells outside of G1 sensitive to YCS conditions at all? In order to control for this, cultures arrested outside of G1 should be analyzed. Moreover, the difference between G1 and non-G1 cells (labeled S phase without apparent reason) does not become clear in the model in Figure 6. Why should Ogg1/Ntg1/2 be insensitive to actin outside of G1?

To start from the last question: Ogg1 and Ntg1/2 cannot be the crucial target of misregulated actin, because we detect YCS in their absence in G1 phase cells. Note that we provide extensive FACS data and clearly label samples as G1 arrested or as exponential cultures (Suppl figs. 2 and 5). We use a standard pheromone synchronization that is monitored by microscopy (bud formation only occurs in early S) and FACS. The G1 population is about 30-40% of an exponential yeast population on rich media. In explanation of the G1 vs S differences, we explain that AP endonuclease appears to act independently of the glycosylases in G1 (see response to Rev 1) That explains Ogg1/Ntg1&2-independent YCS in G1.

4. Given that the Ogg1/Ntg1/2 are bifunctional enzymes and Ogg1 can produce breaks in the absence of any AP endonucleases in vitro, the requirement for AP endonucleases in vivo seems puzzling. This is not addressed here at all.

AP endonucleases are very efficient at creating ligatable ends, and perhaps help coordinate the sequential repair of adjacent breaks. They are highly regulated molecules, which includes trafficking between cytoplasm and nucleus. The fact that Ogg1 can cleave at 8-oxo-G is not new, and is used here to demonstrate that Zeocin can indeed induce both ss and ds breaks, with DSBs being much more rare.

5. The use of CHEF gels for epistasis analysis appears questionable at best. There are no quantifications, which leaves less than black-and-white results open to interpretation as to whether a partial phenotype is closer to “epistatic” or to “additive”. Moreover, the exposure time and Zeocin concentrations vary from experiment to experiment, which makes it very difficult to compare results from different gels/strains. And the concentration used in vitro was order(s) of magnitude lower than in vivo. Is this due to a lack of uptake by the cells?

We have now quantified all CHEF gels and agree that this helps (and in some cases ruled out) interpretations of additivity or epistasis. We are careful only to compare experiments performed on one day, although all CHEF gels experiments were repeated multiple times (see excel sheet in supplemental material). We acknowledge that Zeocin is a reagent that shows batch to batch variability, and that it can lose activity upon exposure to light and heavy metals (the copper chelated form is inactive, only upon reduction of Cu²⁺ into Cu¹⁺ is it active, which normally occurs with the cell). For this reason, we actually titrate each new batch of Zeocin and use the amount (usually 50 µg /ml but sometimes 75 or 80 µg /ml) that does not cause detectable fragmentation without CMB, and provides comparable fragmentation in its presence. We describe this in Materials and methods.

6. Cell cycle experiments need associated FACS profiles. Included now as Suppl figure 2 and 5

7. The auxin experiments need western blots to control for protein degradation. Included, Suppl fig. 3A.

8. The effect of APN1/2 overexpression should also be verified by CHEF gels (Fig. S4C). This we have included in multiple instances: the impact of APN1/2 overexpression CHEF gels is shown in Suppl Figure 3E, Suppl Figure 6.

Minor issues:

9. How was viability measured (Fig. 2D)? The plate images of AP-delta do not reflect the numbers given below the plates, and they appear inconsistent with Fig. 4C.

Thank you for catching this - we erroneously assembled that figure with one wrong drop assay plate. The colony outgrowth is monitored in at least triplicate serial dilution drop assays usually performed using independent cultures. They are a highly robust means for monitoring of lethality due to DNA damage or other insults (see reply to Rev 1). This data now corrected and we have explained the quantitation of growth in Materials and methods.

10. In vitro interaction assays were done with human proteins – is there a reason why yeast proteins were not tested? In the accompanying manuscript, they show some evidence that the phenomenon may be conserved in human cells, but it would be helpful to generate data for yeast proteins as well.

These repair proteins are quite conserved, and we used the human ones as we did not have the yeast purified proteins. However, now we omit the human protein precipitation assay and include only the recovery of the yeast proteins from a yeast extract with well characterized bicyclic peptides (see RJ Gübeli et al., 2021 DOI: [10.1021/acscchembio.0c00825](https://doi.org/10.1021/acscchembio.0c00825)). This provides evidence for selective binding, given that the yeast extract contains at least 4000 soluble protein species. We recover the abundant known actin binding factors (mass spec data) and detect selective binding of Apn1 (not Mcm3 nor Ogg1 by Western blot), which we feel validates this assay. Nonetheless, our model does not depend on repair factors binding actin: rather we focus on the well-established role of actin-containing nucleosome remodelers like INO80 in DNA polymerase processivity.

11. Fig. 3C: please show both data points, not just the average. Done

Reviewer #3 (Remarks to the Author):

This submission by Shimada et al. follows up on published work by this group showing that combination of loss of TOR signaling and zeocin treatment in budding yeast results in chromosome shattering (YCS). The authors now investigate the role of BER in the process of generating the DSBs that yield YCS. They perform an extensive survey of BER genes implicated at distinct steps of this repair pathway. Overall this is a provocative observation. However, it is not entirely clear to what physiological process YCS might be relevant to. YCS is triggered in highly stressed cells lacking TOR signaling and therefore impaired in multiple physiological pathways. These cells have also limited viability. The conclusions rely heavily on CHEF gels without quantitative analysis of the signals and without analysis of multiple biological replicates (see below for details).

Gels are now quantified and replicates are included in a supplemental excel sheet.

YCS occurs in cells treated with Zeocin but not with other compounds generating oxidative damage, suggesting that a large spectrum of lesions is required to start this process. In addition, TORC2 has to be inhibited. Yeast TORC2 orchestrates a complex stress response that regulates membrane lipids and proteins. The ability to prevent/rescue YCS with a variety of mutations in the BER pathway is interesting and provocative. However, the physiological significance of phenomenon is unclear. Indeed, some mutants rescue fragmentation but not viability, suggesting that it is a dead-end pathway. Moreover, what is the connection with TOR signaling? Is this conserved in higher eukaryotes? Chromosome fragmentation occasionally occurs in human tumor cells followed by rebuilding a scrambled chromosome during chromothripsis; YCS seems to be unrelated to this process.

We elaborate more on the relevance of these observations for mammalian systems in the discussion.

We agree that YCS is unrelated to Chromothripsis.

A major technical problem with the manuscript is that it relies mostly on one experimental approach (CHEF gels) and there is no quantitative assessment of the data. This is a significant issue since: 1) these gels are not immune to variability (see examples below); 2) it leads the authors to employ various descriptors, which provide little information about the actual impact of the tested mutation and: 3) it also leads to over-interpretation of some data (see examples below). CHEF gels can be scanned, as done in the Hurst et al. paper, the signal quantitated and average changes calculated with associated statistical significance.

We have now quantified all the CHEF gels and we have augmented the study with additional data that supports our hypothesis.

Specific points:

Figure 1C: It seems that OGG1 alone induces some nicking in the absence of Zeocin, which complicates the interpretation of the data.

The plasmid has low levels of oxidized bases which gives the low level (< 3-4% background) of Ogg1 activity.

There is some inherent variability between CHEF gels showing the same experiment, emphasizing the need for replicates and quantitation. Compare Figure 2B, lane 17 with Figure S1B, lane 10 (rescue by *ogg1/ntg* mutations). Rescue in Supp Figure 1B is less convincing.

We agree that there is some variability, but the quantitation the gels reinforces our conclusions significantly and we are careful to cite the degree of “resistance” as a quantitative ratio (B/A in a given condition over B/A in WT). Please see replies to reviewers 1 and 2.

Lane 179 and Figure 2C: “almost completely”, please quantitate. Done

More consistency with the experimental design would have made the experiments easier to compare. For example, it is difficult to compare the impact of the double vs. triple mutants *apn1*, *apn2*, *rad1* between Figure S1C (BHS, Zeocin 75uM, 45 and 90 minutes) with Figure S1D (CMB, Zeocin 50uM, 60 minutes). There are more instances in which the authors use CMB or BHS without explanation. Since the Hurst submission indicates that CMB is more efficient at triggering YCS and more specific towards TORC2, why not use CMB for all experiments?

Actually we use highly standardized conditions in this paper, although the exact concentrations of reagents used vary with different batches of reagents because their specific activity varies (this is not our fault – we titrate each new batch of Zeocin for damage and then freeze it in small aliquots that are only defrosted once. Ditto for BHS and CMB). We have now included a figure (Suppl Figure 1) that shows that BHS is between 10 and 20 times more efficient than CMB, yet the drugs show the very same selectivity for Tor2 over Tor1, and do not affect related checkpoint kinases Mec1 and Tel1 (see Shimada et al., Mol Cell 2013). Since a *tor2* active-site mutation confers resistance to both drugs, we are convinced the two chemicals act in the same manner (as shown in Supplemental Figure 1, they are closely related molecules). However, as we only recently obtained BHS, we could not use it for all experiments. The amount of Zeocin used is empirically determined as explained in Materials and Methods. The time window does not play a major role as demonstrated in Shimada et al. 2013, as YCS basically plateaus within 30 min (Shimada et al., 2013).

Figure 3A and 3B. It looks as if the double *pol4/pol32* mutant rescues better than *pol32* alone: quantitation should tell. Lane 224-225: “very minor resistance”: I cannot see a difference between lane 4 and lane 8. Also, in this experiment it seems that zeocin alone triggers some fragmentation: compare lanes 1 and 2 and lanes 5 and 6. Similarly, lane 228 figure 3C: “a minor resistance”.

This is now corrected and substantiated by gel quantification, which allows us to use more quantitative terms to describe our data.

Lanes 240-241, Figure S2B: “almost identical”. It seems that the *pcd1* mutant might have more low molecular weight DNA”.

This is now corrected and substantiated by gel quantification.

Lane 250, Figure 4A: “little or no resistance”. Please quantitate. Done

Figure 2B shows a strong rescue by *ogg1/ntg1/ntg2* triple mutant in exponentially growing cells:

compare lane 9 and 18. In contrast, Figure S3A shows poor rescue in exponentially growing cells, compare lane 4 to lane 8. This raises two important issues: 1) inconsistencies between experiments, which can only be addressed by performing multiple biological replicates and quantification and 2) it does not support the conclusion that there is a difference between G1 arrested and exponentially growing cells (S3A) as there is very little, if any, impact in exponentially growing cells between WT and triple mutant”.

The assays are performed as systematically as possible, thus we can only explain the variability by the efficiency of DNA damage due to batch to batch variation in Zeocin (on the other hand, its mode of action is better defined than that of ionizing radiation). The duplicates and triplicates of experiments are quantified and included in a supplemental excel sheet: As mentioned above, we feel that quantitation is most robust when comparing relative degrees of YCS on assays done in parallel.

Compare second lane of Figure 4A and second lane of Figure 4B: this is another example showing that the impact of zeocin alone varies between experiments”.

See above replies, and note that the copper chelated form of Zeocin is inactive, and only upon reduction of Cu^{2+} into Cu^{1+} is it active (this normally occurs within the cell). The conditions we use do not dramatically increase 8-oxoG levels induced by a given amount of Zeocin in vivo (Suppl Figure 2), thus sensitivity or resistance to YCS is likely due to alterations in repair. This is stated in the paper and since different mutant strains are compared to WT within one assay, we consider the quantitation robust.

Lane 269-270 and Figure 4D: “*cdc9*... generated fragmentation on zeocin even in the absence of TORC2 inhibition”. I assume that this statement is based on the comparison of lane 2 and lane 6 (Figure 4D). These samples look nearly identical, which does not support the conclusion.

We agree – you are right and your interpretation is confirmed by quantitation. There is only ~10% difference between WT and the *cdc9* mutant in the absence of TORC2 inhibition, now mentioned.

To monitor actin-BER interactions, the author elected to use actin-binding peptides described in a preprint. If I understand properly, the rationale to use G- and F-actin specific ligands would be to identify actin-binding proteins that interact specifically with polymerized actin. However, the protein gel shows that all identified proteins bind to both peptides, with the notable exception of cofilin, which should bind to both F- and G-actin but does not interact with the F-actin specific peptide.

We do not think there is any preference for the F- actin probe or the F/G-actin probe, thus the proteins recovered bind both, or at least F-actin. Cofilin is also recovered in F-actin peptide binding although to a lesser extent than in the F/G-actin pull-down. This is clarified in the legend now. Cofilin co migrates on the gel with RFA3, which like RFA1 has preference for the F/G-actin peptide.

With regards to the direct interaction between actin and BER proteins in human cells, the author monitor association in the pellet following high-speed centrifugation. However, polymerized actin filaments could trap other proteins and drag them into the pellet in a non-specific manner, dependent on the concentration of proteins. These experiments do not seem to have been controlled for total protein concentration, a factor that strongly influence the ability of proteins to pellet. Actin-containing samples have a protein concentration several fold higher than the control. It might have made more sense to pull down actin in solution and monitor binding.

See comments to Reviewers 1 and 2 above. We agree that the human spinout assay is fraught with artefact and only include the yeast bicyclic peptide-actin recovery assay. Robustness is explained above.

Our model does not depend on actin-repair enzyme interaction, as we have evidence that actin-dependent remodelers have similar effects on YCS.

Typos..

Lane 57: one-ended parenthesis. Thank you - fixed

Lane 247: one-ended parenthesis. Thank you - fixed

Lane 251: remove "cf". Thank you - fixed

Lanes 258-259: please check lane 258. Also, this should be rewritten in a way that does not suggest pol4 mutant had an impact on exponentially growing cells

You are right – this is now corrected

Response to REVIEWER COMMENTS Nature Communications ms NCOMMS-20-19157A

Shimada et al

Reviewer #1 (Remarks to the Author):

In the revised manuscript, Shimada et al. have successfully addressed many of my concerns/comments. Unfortunately, there are still issues with the new version, particularly with the *quantitation methods they have used*. In the following list, I have gone through the rebuttal comment by comment:

1)A major concern with this work is the lack of quantitation of SBs from the gels for different cell strains (e.g., wt vs. mutant) or under different conditions.

(p.24, 1st par.) As implied by the authors, the photo intensities are linear only over a small range and don't correlate with the actual pixel intensities of the scans. However, the Typhoon FLA 9500 response is linear over 5 orders of magnitude and Bio Rad chemiDoc XRS system has a dynamic range of 4 orders of magnitude. Therefore, one doesn't have to worry about linearity over the changing gel scan area due to changes in exposure.

We have clarified our quantitation methodology and we believe that the results we present are linear, robust and reliable. Because we were not sufficiently clear in our previous ms, there are several misunderstandings that led to the reviewer's comments. We have now provided extensive description of the gel conditions, the quantification methodology and have added figures that allow the reader to follow our arguments about the number of DSBs incurred, and the B/A quantitation. Relevant details are in Materials and Methods and the Supplemental Figure 2C-D. See also Figs 1 & 2 for reviewers at the end of this rebuttal.

To answer the points made above: first, we are not scanning photographs. We capture the fluorescence on the generally on the Typhoon FLA 9500 scanner with LPB (510LP) filter (GE Healthcare) or in early stages, with the Chem Doc XRS system (Bio-Rad). Both show a linearity of dsDNA detection over 10'000 fold dilution (see Figure 2 for reviewers below). The gel data were then directly transferred into Image J software.

For the quantitation, the rectangle which covers an entire lane was set as ROI, and DNA intensity was plotted. Background from a flanking region of the gel without sample was subtracted. We found no difference in monitoring intensity in Image J and the Typhoon scanner program, ImageQuant (see Figure 1 for reviewers). In the Image J program, a region of interest (ROI) spanning from above the largest chromosome band (2Mb) to ~20 kb (determined by size markers in the gel) was created and the signal intensity was measured using line plot profiling. The same ROI was applied for each lane in a CHEF gel image (note: in rare cases where the gels did not run straight, we adopted the ROI boxes appropriately. Note that because B/A is an "internal lane" ratio, this is still valid.

To measure B/A value, a division of the lane above and below 0.57 Mbp was created (above 0.57 Mbp includes Chr VIII/V bands through the largest Chr IV/XII; below extends from 0.56 Mbp (*below* Chr VIII/V) to about 20 kb, and includes most chromosome fragments and the small Chr I, III, VI, and IX). Whether or not there are ss gaps or nicks in the chromosome fragments is not relevant, nor is it scorable with nondenaturing gels. The intensity of ds DNA above (A) and below (B) 0.57 Mbp was measured. As long as the gels ran straight, the same ROI was shifted horizontally to measure all lanes in the gel. B over A was calculated and is indicated under each lane of each CHEF gel, representing the degree of

chromosome fragmentation. We are convinced that the B/A value comparison is robust and monitors irreparable DSBs which is what we aim to monitor.

We also modeled B/A ratios based on the mean fragment size detected by CHEF gels, and the values (see Supplemental Figure 2D) fit our measurements well. For B/A in the range of the experimental values (i.e., 0.2 – 13), the model predicts a mean fragment size of 136-624 kb (see plot in Supplemental Figure 2D). Note that the slope of the B/A curve around the mean fragment size of 100kb is very steep, so for mean fragment size of 100kb the predicted B/A is 50. We agree that mean fragment size is difficult to monitor if fragmentation is not complete, and therefore in the text we removed all references implying “x-fold” differences calculated by comparing B/A from different lanes, as requested.

Unfortunately, since the ImageJ program was used for quantitation of gel images, I assume screen shots of the data were analyzed. *This assumption is incorrect.* Because of the reason stated above, the analysis is more accurate when the file export information is directly analyzed. *This is what we did.* For the Typhoon data, this would be with the ImageQuant program. As stated below, the key is to obtain the number-average length (L_n).

We have the length of line data for all gels, but rigorous comparison showed that this was less robust than our method of integration of intensity within a ROI. We have systematically compared our method with others and are convinced this is the most reproducible method for fluorescent gel quantitation of “smears” of dsDNA created by random DSBs.

2)Indeed, this might reveal a critical ‘break frequency’ that is needed to generate YCS.

YCS arises from misregulation of BER, and we estimated the number of DSB incurred by 50-80 $\mu\text{g/ml}$ Zeocin (average amount used) to lie between 80-140 (for an average terminal ds fragment size of 100 kb \pm 20 we calculate 112 ± 22 DSB), in wild-type cells in the presence of TORC2 inhibitors. The number of ss lesions converted to DSBs depends on Zeocin concentration used (Shimada et al., 2013). We had already included modeling data that estimates the number of DSBs incurred to reduce the chromosomal complement to an average ds fragment size of a given mass (Shimada et al., Figures 1C and D). We now elaborate on this, discussing it in detail. We also extend the modeling for average fragment sizes from 70 to 700 kb in **Suppl Figure 2C** (see below). Note that the overall base oxidation frequency does not change with TORC2 inhibition (**Suppl Figure 2A**), but the *conversion* of ss lesions to DSBs does. Without TORC2 inhibition the levels of ss lesions induced are readily repaired.

(p.24, 2nd par.) The ratio chosen (B/A) is systematically flawed because the A region (>0.57 Mbp) contains long SSB fragments, and the amount changes with dose. Thus, this ratio would give a nonlinear dose-response (SSB/ μg vs [Zeocin]).

We are not monitoring ssDNA nor ss breaks, which would require a different staining reagent and probably alkaline gels. We use neutral *nondenaturing* gels and we cannot detect ssDNA as we use dyes that intercalate dsDNA. According to extensive literature, EthBr and SYBR safe dye (Invitrogen) are at least 10 fold more sensitive to ds than to ss DNA. Within a given experiment (with fixed levels of Zeocin) the number of **abasic sites** (resulting from glycosylase activity) is constant and this does not alter upon TORC2 inhibition (**Suppl Figure 2A**). In other words, we are exclusively monitoring the conversion of Zeocin induced base oxidation into DSB resulting from TORC2 inhibition, which occurs more readily after Zeocin or Bleomycin treatment, because these (closely related) reagents have a tendency to generate

“paired” or clustered oxidation events (Povirk 1996). Our goal was to score the relative rate of DSB generation from a fixed amount of Zeocin and determine how these arise. Povirk (1996) showed that bleomycin-like reagents (including Zeocin) produce ss nicks and ds breaks in a 10:1 ratio. Thus, it is reasonable to assume that the ss nicks are 10 times as abundant as the DSB numbers we score.

While we agree that DNA fragments carrying long-stretches of ssDNA might migrate aberrantly and more slowly, we also note that large ds chromosomal fragments that contain extensive ss stretches will be trapped in the well during PFGE (c.f. behavior during S phase, or upon HU arrest). Note that under our conditions (Zeocin \pm CMB) we do not observe an increase in non-migrating high molecular weight DNA. This argues that chromosomes with ss nicks probably are migrating as intact chromosomes do, which is expected under neutral agarose (salt-containing) electrophoresis conditions.

The following text is now included to explain how we can model break formation from resulting average fragment size:

“Quantification of the mean number of DSB per chromosome was determined based on the assumption that DSBs are independently and uniformly distributed, i.e. they occur at the same frequency (λ) per unit length of DNA in all the chromosomes. Thus the number of breaks in chromosome i follows a Poisson distribution of parameter λS_i , where S_i is the size of chromosome i . In this model, the mean fragment size over all the chromosomes is $S_{tot}/(\lambda S_{tot}+16)$, where S_{tot} is the length of the genome. Hence the mean number of breaks per chromosome can be calculated as a function of the mean fragment size. Given an estimated mean fragment size in the YCS experiments (i.e. maximal intensity distribution curve) of 100 ± 20 kb, the corresponding mean number of breaks per chromosome are from Chr I to XVI are: 1.96; 6.92; 2.69; 13.04; 4.91; 2.30; 9.29; 4.79; 3.74; 6.35; 5.68; 9.18; 7.87; 6.68; 9.29; 8.07 (or 112 ± 22 total breaks; Fig. 1C-D). An extension of the graph in Figure 1C to an average fragment size of 700 kb is shown in **Supplementary Figure 2C**. A mean fragment size of 100-300 kb corresponds to a mean number of DSB genome-wide of about 25 (for 300kb average) to 110 (for 100kb average). Note that this is based exclusively on dsDNA detection (as fragments as well as full length chromosomes), on neutral gels that do not denature DNA. “

There are several papers that address the quantitation of these types of gels [e.g., Czaja et al. DNA Repair 9(9):976, 2010; Li et al. Sci. Reports 11(1):18393, 2021]. The background of undamaged chromosomes on CHEF gels can be subtracted to give a 'smear' which allows for determination of the ensemble average, the median length, & the number average length (L_n). It follows that $SSBs/(\text{unit DNA}) = 1/L_n (+Zeo) - 1/L_n (-Zeo)$.

Thank you for the suggestion. However, the Li et al paper is all alkaline gel assay and in the Czaja paper, all but one gel are alkaline denaturing gels, unlike our analysis.

3) the authors should use it to quantify the number of SSB vs. DSB from these gels for accurate comparisons of the different agents.

Thank you for the suggestion. However, as discussed above, using neutral CHEF conditions does not allow us to distinguish full length chromosomes from those with small internal stretches of ssDNA, or with ss nicks. In any case, what we monitor is *the relative rate of conversion of ss lesions to DSBs*, given a fixed level of base oxidation by a set Zeocin concentration in a given experiment. There is some variation in the amount of DNA loaded in each lane of a CHEF gel, as we are apply an agarose plug, which is why we decided to quantify using internal measurements of “small fragments <560 kb” vs “intact

chromosomes >570 kb". This yields a ratio specific for each lane, which gives a robust measure of DSB frequency, and the conversion of base oxidation to DSB.

It is important to note that the TORC2 inhibitor *does not increase the rate of oxidative lesions incurred* (abasic site frequency for a given amount of Zeocin is unchanged by TORC2 inhibition, **Supplemental Figure 2A**). Povirk (1996) estimated that bleomycin induces 10 times more ss nicks than DSB. We find no reason to challenge this. Interestingly, the smallest average size ds fragment (limit) we detect is in the *cdc9* mutant at nonpermissive temperature, which produces an average fragment size around 20kb, or roughly 600 DSBs per genome. We assume that loss of ligase 1 (*cdc9*) likely blocks all types of repair, (except that mediated by Lig4, which was not affected by TORC2 inhibition, Shimada et al 2013). More frequently we detect an average fragment size of 100 ± 20 kb, which represents a range of 112 ± 22 DSBs in wild-type cells. Obviously, most oxidized bases are not generating DSBs; yet it only takes a few irreparable DSBs to kill a cell (see survival plots after transient exposure to YCS conditions in Figure 4C,D)

4) authors should quantify the differences in overall strand breaks for different conditions; at least the percentage of breaks should be computed.

It is unclear to us what is meant here by "percentage of breaks". Percentage of DSB over ss lesions ? or over oxidized bases ? We believe this has been answered above (in wt cells it is 1:10; Povirk 1996).

5) P. 10 (line 228): Again, requires quantitation to make this statement.

We agree that one cannot convert B/A ratios into "absolute frequencies" of breaks, but one can calculate a B/A ratio graph that correlates with an average final product size (see panel B below). We removed all statements about "fold change" based on B/A ratios. We show in **Supplemental Figure 2D** the calculated B/A ratio for each value of λ (mean fragment size over all chromosomes ranging from 70 – 700 kb; Panel B) as follows:

A

B

Figure legend. **A**, For a given mean fragment size, we calculate the mean number of breaks for each chromosome and pile those numbers from chromosome 1 in the bottom to chromosome 16 in the top. Hence, the solid red line represents the mean number of breaks in the genome. **B**, B/A ratios from the model as a function of mean fragment size over all the chromosomes.

In the model, we assume that DSBs are independently and uniformly distributed, i.e. they occur at the same frequency (λ) per unit length of DNA in all the chromosomes (Cedervall and Källman 1994 – now

cited in the paper). Under these assumptions, the DSBs locations are described by a Poisson process with parameter λ . Thus, the number of breaks in chromosome i follows a Poisson distribution of parameter λS_i , where S_i is the size of chromosome i . In this model, the mean fragment size over all the chromosomes is $\frac{S_{tot}}{\lambda S_{tot} + 16}$, where S_{tot} is the length of the genome. Hence the mean number of breaks per chromosome can be calculated as a function of the mean fragment size (see panel A).

The theoretical relative intensity distribution, $I(x)$, of DNA fragment from randomly distributed DSBs is expressed as

$$I(x) = \lambda x e^{-\lambda x} (32 + S_{tot} \lambda - 16 \lambda x)$$

(see eqn. 2 in Cedervall and Källman 1994).

Therefore, the theoretical ratio B/A for e.g. A > 550 kb and B < 550 kb would be given by

$$\frac{\int_0^{550000} I(x) dx}{\int_{550000}^{\infty} I(x) dx}$$

Cedervall, B., and Källman, P. "Randomly Distributed DNA Double-Strand Breaks as Measured by Pulsed Field Gel Electrophoresis: A Series of Explanatory Calculations." *Radiation and Environmental Biophysics* 33, no. 1 (March 1994): 9–21.

For most of our B/A values (ranging from below 1 for intact chromosomes to values between 1 and 50) the mean fragment sizes range from 150-300 kb. Based on this graph, these B/A values estimates a from 80 to 20 DSBs genome-wide.

6) P. 12 (lines 260-261): Again, quantify.

Text has been appropriately correctly; all data are quantified.

7) P. 13 (lines 281-283): Could it be that DNA pol and/or RNA pol elongation blockage induce enhanced AP endo activity (i.e. G1 vs log-phase)?

We think this is unlikely because we see efficient conversion to DSBs in G1 (no DNA replication) and the addition of α -amanitin, which inhibits transcription does not mimic TORC2 inhibition (Shimada et al., 2013). We also observed that the acute Rpo21 (Rpb1, PolII catalytic subunit) degradation did not enhance DSB formation (not shown)

8) The more compact chromatin (on average) in G1-phase cells may partially 'release' BER enzymes from the 'hand-off' mechanism (e.g., Prasad et al., JBC, 2010) and APE activity may become more active.

We do not know of any data showing that chromatin is systematically more compact in G1 unless cells enter stationary phase, which is not the case here. But we do discuss that Apn1 activity varies with stages of the cell cycle (see Discussion) and show that chromatin accessibility changes with TORC2 inhibition.

It is confusing to tell if P/T panel goes with 7B or 7D. This should be clarified.

Corrected : the P/T panel is labeled independently.

9) Quantitation of the number of SB induced by Zeocin, \pm CMB, would address this possibility.

The number of abasic sites induced by a given amount of Zeocin is the same \pm CMB (Suppl Figure 2A).

Minor Comments

All OK Thank you

Minor Comments on new version:

p. 6 (lines 153-153): The statement "...nor is there any sign of DSB formation..." doesn't align with the data. Clarify. Removed

p. 17 (line 490): Delete "nuclei" actually the word "in" was missing. The phrase now reads "in nuclei" ...

p. 17 (line 496): ...subunit of [the] INO80 complex, which [is] an actin nucleosome..... Corrected

p. 24 (lines 720-722): Sentence is not clear and should be rewritten. Right – this is now corrected

Reviewer #2 (Remarks to the Author):

In their revised version, the authors have addressed many of the reviewers' original concerns and have strengthened their arguments substantially. This has resulted in a more coherent and convincing study.

The effect of Zeocin on DSB formation via the BER pathway is now very well documented.

The link to the actin system via the INO80 complex is again plausible, but still speculative because relevant functional data (e.g. effects of *ino80* mutants on YCS) are not provided. Such data appear to be in revision in an additional manuscript (Hurst et al) that the authors mention but do not show; including them in this manuscript would have helped the authors to make their point. Nevertheless, I am mostly happy with how the authors have responded to the reviewers' comments.

The inclusion of INO80 data in Shimada et al. would require first demonstrating that *in vivo* there is more G-actin in the nucleus. This demonstration is contained in 4 figures in Hurst et al. , which show that TORC2 inhibition and/or Las17 degradation drives higher nuclear G-actin levels. Without this demonstration, there is no proof that INO80C or other remodelers would be affected, and therefore the *ino80* mutant data fits better in the Hurst et al paper. We also note that the Hurst et al paper rules out other modes of action for nuclear actin, and that we test a range of nucleosome remodelers. These important experiments would not fit in a merged paper.

For information only: Rebuttal for **REVIEWER COMMENTS**
Nature Communications manuscript NCOMMS-20-19089A

Hurst, Gerhold et al

Reviewer #1 (Remarks to the Author):

The authors have made a number of revisions to their manuscript. Two points in particular have been addressed: first, the quantification of their CHEF gels strengthens the quantitative YCS measurements overall. Second, they show a link of the YCS phenomenon to the actin-containing chromatin remodeler INO80.

This latter finding specifically strengthens the model that the authors put forth in the accompanying manuscript (Shimada et al). In my review of that manuscript, I mentioned that inclusion of the INO80 data would significantly enhance the impact of that study. Evaluation of this present manuscript further confirms my view on the pair of manuscript: I would strongly recommend moving the INO80 data into the Shimada et al manuscript; the resulting study would be a very nice addition to the journal.

In contrast, the Hurst et al. manuscript does not meet my expectations of a well-rounded mechanistic study in this journal. My previous impression (an extensive collection of relatively loosely connected data that lacks a stringent logical flow) remains unchanged, despite the authors' detailed explanations in response to my concerns. I still do not understand the relevance of their phosphoproteomics analysis to their conclusions. The section on mammalian cells remains isolated and without much mechanistic insight other than that the mechanism likely differs from the yeast situation. I do not wish to argue that the data shown here are not valuable, but I do not think that the format in which they are presented lends itself to a publication in this journal.

The rigorously documented take-home from the Hurst et al manuscript is that there is a change in G-/F-actin balance upon TORC2 inhibition (and/or Las17 ablation), and that this interferes with oxidized base repair through nucleosome remodelers. The reason this is very important to the field is because the following papers (many of which were in *Nature* journals) claim that filamentous actin in the nucleus has either a positive role in DSB repair, or acts at stalled replication forks. We show that nuclear actin may increase remodeler activity and that it is deleterious to repair. These are recent papers relevant to the question of actin in repair.

Zagelbaum et al., 2023 *Nat Struct Mol Biol.* 2023 30(1):99-106. doi: 10.1038/s41594-022-00893-6

Palumbieri et al. 2023 *Nat Commun* 2023 1:1 p 7819 DOI: 10.1038/s41467-023-43183-5

Caridi et al., *Nature* 2018 559 :7712 p 54-60 DOI: 10.1038/s41586-018-0242-8

Schrank et al., *Nature* 2018 559: 7712 p 61-66 DOI: 10.1038/s41586-018-0237-5

Nieminuszczy et al., 2023 *Nucleic Acids Res* 51:12 p 6337-6354 DOI: 10.1093/nar/gkad369

Han et al., *Nature Commun* 2022 13; 3743; DOI: 10.1038/s41467-022-31415-z

Along with papers on the positioning of DSB (or chromatin context) being important for repair such as Chen et al. *Nat Cell Biol* 2023, 25:1384

Schep et al *Molecular Cell* 2021 81(10), 2216–2230.e10. doi.org/10.1016/j.molcel.2021.03.032

Mitrensi et al., *Molecular Cell* 2022, 82:2132-2147 e2136.

see also : Belin et al., *E LIFE* 2015; Lamm et al., *NCB* 2020

The first 6 papers argue that actin and its chaperones (WASP and ARP2/3) generate nuclear actin filaments in the nucleus that are essential for damage clustering or processing (or in Caridi et al., dynamics), and that these guide DSB repair or replication fork stability. Later papers stress the

pathology of nuclear actin and 3D subnuclear clustering of damage. Our paper shows that in yeast WASP deletion generates DSB from ss lesions, but not due to the absence of nuclear F-actin nucleation by WASP and ARP2/3 at damage, but by upregulating actin-containing nucleosome remodelers due to enhanced accumulation of nuclear G-actin. This is an important alternative explanation of many of the data presented in these papers, and justifies a highly visible publication of Hurst et al.

The importance of the phosphoproteomics screen is that it identifies the effectors of the Ypk1/Ypk2 inhibition, triggered by TORC2 inhibitors. The most significant GO term for proteins with YCS-specific phosphorylation were linked to control of the cytoplasmic actin network and a specific set of complexes at the plasma membrane. Whereas other papers have implicated nuclear F-actin filament formation in DSB repair or replication fork stability, we show that the ablation of an actin chaperone does alter G-/F-actin balance and leads to altered BER, independently of DSB repair pathways and S-phase replication.

Reviewer #2 (Remarks to the Author):

The revised manuscript by Hurst et al. have successfully addressed most of my concerns/comments. Unfortunately, there are still issues with the quantitation methods they have used. I have addressed these new concerns with my responses to the rebuttal letter below:

1) P. 5 (lines 111-112): What about in the presence of Zeocin (i.e. increases cell permeability)? Authors should discuss that possible increased permeability was ruled out in Shimada et al., 2013 (tor2-V2126G mutant) and accompanying manuscript with glycosylase and APE1 mutants.

We now include this argument, and above all, we stress that abasic site generation is not enhanced by TORC2 inhibition. This would not be the case if TORC2 inhibition increased Zeocin uptake or activity.

2) Next 3 comments: P. 7 (lines 149-153): These types of comparisons invoke the need for quantitation of the amounts of DSBs in each case (not simply 'more' or 'less'). See comments in accompanying manuscript. Now provided, see changes throughout the ms's about comparative levels of DSBs.

P. 9 (lines 206-210 and lines 213-215): Again, these differences require quantitation. Now provided. P. 17 (lines 421-422): Again, quantitation of the DSBs makes this data much stronger; gel scans give a 'qualitative' picture and are not a linear representation of the signals. Now provided.

- As detailed in the review of the accompanying manuscript, the B/A ratio is flawed by systematic error (as opposed to statistical error), and the SSB/(unit DNA) should be calculated from the ensemble average of the corrected gel scans.

We believe this point arises from a misunderstanding of our gel system: we are not monitoring ss DNA nor ss fragments, only dsDNA and the average size of subchromosomal fragments. We refer to the comments about quantitation in Shimada et al, and we make it clear in the paper that we are only monitoring DSBs, not ss DNA accumulation nor SSB frequency (which as mentioned above may be up to 10 times the DSB frequency).

- These numbers will provide an accurate account of the gel data and match the high quality of the rest of the manuscript.

We hope that the quantitation we have provided on the frequency of DSBs has clarified the issue.

3) All Minor Comments:
corrected

Figures for reviewers on gel quantitation:

Fig 1 Imagequant vs Image J

ImageQuant and ImageJ quantification B/A value comparison of SYBR safe stained CHEF gels.

Log2 scale

Fig 2 Linearity of SYBR safe dye and Typhoon scanning (our main tool). A two-fold dilution series of GeneRuler 1 kb DNA Ladder (Thermo Fisher Scientific) from 20 to 0.00975 ug of total ladder DNA were loaded on 1 % agarose gel and run in 1 x TAE buffer 100 V for 1h. The DNA was stained by SYBR Safe dye (Thermo Fisher Scientific) and scanned by Typhoon FLA9500. DNA bands were quantified by ImageQuant software and the intensity (arbitrary unit) was plotted over the DNA quantity (ng). We typically apply 95 - 240 ng total genomic DNA (0.8 - 2 E7 cells equivalent) in one lane for CHEF gel analysis, which corresponds to 2.1 - 30 ng of chromosomal DNA.